# PERIODIC SET TRANSFORMER: MATERIAL PROPERTY PREDICTION FROM CONTINUOUS ISOMETRY INVARIANTS

## ABSTRACT

Material or crystal property prediction using machine learning has grown popular in recent years as it provides an accurate and computationally efficient replacement to classical simulation methods. A crucial first step for any of these algorithms is the representation used for a periodic crystal. While similar objects like molecules and proteins have a fixed number of atoms and their representation can be built based upon a finite point cloud interpretation, periodic crystals are unbounded in size, making their representation more challenging. In the present work, we adapt the Pointwise Distance Distribution (PDD), a continuous isometry invariant for periodic point sets, as a representation for our learning algorithm. While the PDD is effective in distinguishing periodic point sets up to isometry, there is no consideration for the composition of the underlying material. We develop a transformer model with a modified self-attention mechanism that can utilize the PDD and incorporate compositional information via a spatial encoding method. This model is tested thoroughly with and without the use of compositional information on a variety of crystal datasets including the commonly used crystals of the Materials Project.

## 1 INTRODUCTION

A crystalline structure is made up of a repeating arrangement of atoms. Crystals can distinguish themselves by both the species of atoms they contain as well as how these atoms are structured. Both of these aspects can determine the various properties of a crystal. Knowledge of these properties is pertinent for determining whether a crystal can be experimentally synthesized or is useful for a particular application.

Determination of property values can be done using ab initio calculations with techniques like density functional theory (DFT) (Sholl & Steckel, 2011) or force field levels (Niketic & Rasmussen, 2012). These techniques can vary in accuracy and are often computationally expensive (Cohen et al., 2012). In some cases, prohibitively so. Further, they require extensive domain knowledge to be applied to even a single crystal, making them inaccessible. This has resulted in a search for alternative methods. In recent years, machine learning has become very popular for this task and has experienced success in decreasing computational costs while producing accurate predictions.

Any application of machine learning requires the adaptation of a data representation that adequately describes the object of interest and is compatible with the algorithm being used. Objects similar to crystals, like molecules and proteins, are often treated as finite point clouds. This makes their representation more easily constructible than a representation for crystals, which are not bounded in size. Further, while a crystal can be described in several ways, descriptors that are easily human-interpretable, such as unit cell parameters or atomic coordinates are not ideal for machine learning algorithms. Atomic coordinates, for instance, do not retain invariance under rigid motion. Unit cell based descriptors are also ambiguous as there are an infinite number of valid unit cells for any given periodic crystal.

The *structure-property relationship* (Le et al., 2012) dictates that changes in the structure of a material result in changes in its properties. Distinction between crystals then allows for distinction between their respective property values. Fundamentally, a machine learning algorithm (for a re-

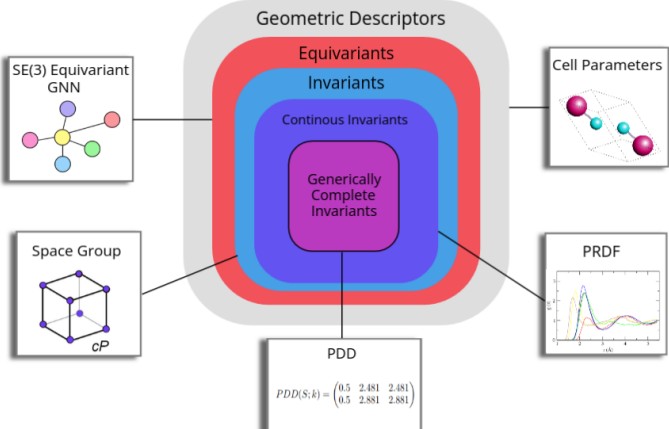

Figure 1: Classification of geometric descriptors for periodic crystals based on the properties possessed.

gression task) is a mapping from representation to value. If a representation cannot *distinguish* periodic crystals then two different crystals can incorrectly be perceived to be the same and so will the output property values. Similarly, if the same crystal can be represented in different ways, mapping to the same value cannot be guaranteed.

The ideal crystal representation is an isometry invariant. Crystals are best treated as periodic point sets which are unordered and span infinitely Smith & Kurlin (2022). Two periodic point sets $\mathbb{S}$ and $\mathbb{Q}$ are *isometric*, if there exist bijective isometries $f : \mathbb{S} \to \mathbb{Q}$ and $g : \mathbb{Q} \to \mathbb{S}$ such that $f(\mathbb{S}) = \mathbb{Q}$ and $g(\mathbb{Q}) = \mathbb{S}$. An *isometry* $f$ is a distance preserving map between metric spaces of the form $d(\boldsymbol{s}_i, \boldsymbol{s}_j) = d(f(\boldsymbol{s}_i), f(\boldsymbol{s}_j))$ for any $\boldsymbol{s}_i, \boldsymbol{s}_j \in \mathbb{S}$ and valid metric $d$. Since crystals are treated as point clouds these isometries include those that fall under rigid motion: translation, rotation, and reflection. An invariant $I$ is a representation that does not change under the application of these isometries. Not all invariants can be considered equally useful, however. Space groups, for example, reflect the symmetry of a given material but tiny changes in atomic positioning caused by perturbations can change the material's space group. While this is enough to distinguish two crystals, no indication is given of *how* different they are. Figure 1 illustrates the relationship between geometric descriptors based on their properties. The ideal invariant is one which possesses the following qualities (Widdowson & Kurlin, 2022, Problem 1.1)

1. *Invariance*: If two crystals (or periodic sets) $\mathbb{S}$ and $\mathbb{Q}$ are isometric, then $I(\mathbb{S}) = I(\mathbb{Q})$.

2. *Completeness*: If $I(S) = I(Q)$, then the two crystals are isometric.

3. *Metric*: A distance function $d$ on $I$ satisfies the following: 1) $d(I(S), I(Q)) = 0$ iff $I(S) = I(Q)$. 2) $d(I(S), I(Q)) = d(I(Q), I(S))$. 3) For any three crystals $S, Q$, and $W, d(I(S), I(Q)) + d(I(Q), I(W)) \geq d(I(S), I(W))$

4. *Lipschitz continuity*: If a periodic set $Q$ is obtained by shifting points within periodic set $S$ by at most $\epsilon$, then the distance between the two periodic sets can be bound according to $d(I(S), I(Q)) \leq C\epsilon$ for some fixed constant $C$.

5. *Computability*: Construction and comparison using the invariant $I$ can be done efficiently.

In the present work, we will use an isometry invariant called the Pointwise Distance Distribution (PDD) which has these qualities (Widdowson & Kurlin, 2022). The PDD, defined formally in Definition 3.2, is invariant, generically complete, and has an established continuous metric using the Earth Mover's Distance (Rubner et al., 2000). It can also be constructed in near-linear time, making its application to large datasets feasible.

The main contribution of this work is a model based on the transformer architecture (Vaswani et al., 2017) which utilizes the PDD of a crystal to make predictions on the properties of materials. In doing this, we bridge the gap between unambiguous crystal descriptors and machine learning models.

We show that such a representation is effective in producing results that are on par or better than graph-based models, despite the additional structuring of data that comes in the form of edges and edge embeddings. Finally, we develop a method called *PDD Encoding* that enables the PDD to be combined with compositional information for improved performance. Our application is divided into two parts; the first considers the determination of lattice energy from structure alone, while the second uses both structural and compositional information to predict properties for the crystals in the Materials Project.

## 2 RELATED WORK

Early works in crystal property prediction used more classical statistical methods like kernel regression (Calfa & Kitchin, 2016) before eventually moving towards deep learning (Ye et al., 2018). More recent works have shifted to Graph Neural Networks (GNN) (Xie & Grossman, 2018; Choudhary & DeCost, 2021; Yan et al., 2022; Omee et al., 2022; Park & Wolverton, 2020; Chen et al., 2019; Das et al., 2022; Cheng et al., 2021; Sanyal et al., 2018; Schütt et al., 2018) due to their ability to make use of structured data. Several of these focus on predicting the properties of the crystals contained within the Materials Project (Jain et al., 2013) using a multigraph representation where vertices represent atoms and edges are embedded with the pairwise distances to an atom's nearest neighbors. Some state-of-the-art models use line graphs to incorporate more geometric information like angles and dihedrals (Choudhary & DeCost, 2021; Ruff et al., 2023). The derived line graphs can contain significantly more vertices and edges, incurring a higher computational cost. Lin et al. (2023) take a physics principled approach and substitute the interatomic distances for interatomic potentials and capture a crystal's periodicity using the infinite sum of these potentials.

While graphs are effective in modeling crystal structures, they are discontinuous under perturbations (Edelsbrunner et al., 2021). Small movements in the atomic positioning can cause significant changes to the graph's topology. Some graphs also rely on the choice of unit cell. Due to an infinite number of possible unit cells, the graph is then reliant on the data or the cell reduction technique used.

Equivariant (Thomas et al., 2018) models are a step in the right direction when compared to those that rely on non-invariant geometric descriptors. Even so, invariance is a stronger and thus, more desirable property than equivariance. Invariant representations are required for distinguishing between crystals, while equivariant representations cannot be relied upon to do so. Equivariant transformers have been developed for finite points clouds (Fuchs et al., 2020), but the periodic case has yet to be handled. Equivariant GNNs are still limited by discontinuity and reliance on the unit cell Du et al. (2022); Xie et al. (2021).

Many invariants for crystal structures exist, but few with all the desired properties. Smooth Overlapped Atomic Positions (Bartók et al., 2013) and Atomic Cluster Expansion (Drautz, 2019) for example, are both invariants, but due to the reliance on a finite subset of the periodic set, lack continuity. In addition to the properties mentioned earlier, the invariant needs to be able to be used in a machine learning algorithm. Further, it needs a way to incorporate compositional information as invariants typically only consider structure. Some invariants have been adapted for use in machine learning algorithms such as symmetry functions (Behler, 2011; Egorova et al., 2020) and Voronoi cells (Ward et al., 2017). Both of these, however, still lack continuity. The Partial Radial Distribution Function is invariant and continuous but is not complete for homometric crystals. Average Minimum Distance (AMD) (Widdowson et al., 2022) is invariant and continuous and has been used to predict lattice energies via Gaussian Process Regression (Ropers et al., 2022), but does not have a way to incorporate compositional information.

## 3 METHODOLOGY

We can define a crystal more generally in terms of a periodic point set (Smith & Kurlin, 2022) or periodic set for short.

**Definition 3.1** (Periodic Point Set). *For a set of $n$ basis vectors $\boldsymbol{v}_1 \ldots \boldsymbol{v}_n \in \mathbb{R}^n$, the lattice $\mathbb{L}$ is formed by the integer linear combinations of these basis vectors $\{\sum_{i=1}^{n} c_i \boldsymbol{v}_i | c_i \in \mathbb{Z}\}$. The unit cell is the space spanned by the parallelepiped $\mathbb{U} = \{\sum_{i=1}^{n} t_i \boldsymbol{v}_i | t_i \in [0, 1)\}$. For a unit cell $\mathbb{U}$ in the*

*lattice* $\mathbb{L} \subset \mathbb{R}^n$, *the motif* $\mathbb{M}$ *is a finite subset of* $\mathbb{U}$. *Then, a periodic point set* $\mathbb{S}$ *of lattice* $\mathbb{L}$ *and motif* $\mathbb{M}$ *is defined by* $\{\boldsymbol{\lambda} + \boldsymbol{p} : \boldsymbol{\lambda} \in \mathbb{L}, \boldsymbol{p} \in \mathbb{M}\}$.

The PDD of a periodic set is the $m \times (k+1)$ matrix where $m$ is the number of atoms in the motif $\mathbb{M}$ and $k$ is a positive integer indicating the number of nearest neighbors to use. Each row corresponds to a point in the motif and the entries within the row consist of the Euclidean distance to each of this point's $k$-nearest neighbors within the entire periodic set $\mathbb{S}$. The first entry of the row is assigned to be a weight equal to $\frac{1}{m}$ (the distances follow). Once the matrix is formed, rows that are the same are collapsed into a single row and their respective weights are added. Due to very small differences between rows caused by floating point arithmetic or atomic perturbations, it is common to use a tolerance, henceforth called the *collapse tolerance*, that allows rows with small non-zero differences (e.g. with respect to $L_\infty$ distance) to be treated as the same. By collapsing rows in the PDD, the resulting matrix representation is always the same for a given crystal, regardless of the unit cell. Formally,

**Definition 3.2** (Pointwise Distance Distribution). *For a periodic set* $\mathbb{S} = \mathbb{L} + \mathbb{M}$ *with a set of motif points* $\mathbb{M} = \{\boldsymbol{p}_1, \dots, \boldsymbol{p}_m\}$ *within a unit cell* $\mathbb{U}$ *of lattice* $\mathbb{L}$, *the PDD matrix for a parameter* $k \in \mathbb{N}^+$ *is a* $m \times (k+1)$ *matrix where the* $i^{th}$ *row consists of the row weight* $w_i = \frac{1}{m}$ *followed by the euclidean distances* $d_1 \dots d_k$ *from the point* $\boldsymbol{p}_i$ *to its* $k$-*nearest neighbors such that* $d_1 \leq d_2 \dots \leq d_k$. *If a group of rows is found to be identical (or close enough using a valid distance measure within some tolerance) then the matrix rows are collapsed and the weights of the involved rows are summed. The rows of the matrix are finally ordered lexicographically.*

### 3.1 PERIODIC SET TRANSFORMER

In our model, rather than being considered a matrix of values, the PDD will be considered a set of grouped atoms. A single group of atoms corresponds to the $k$-nearest neighbor distances in a given row within the PDD matrix. Each member of the set will carry the weight provided by the row in the PDD. Any set $\mathbb{A}$ can trivially be turned into a weighted set by weighing each element by $\frac{1}{|\mathbb{A}|}$. When the PDD is not collapsed, then there can be more than a single occurrence of any given element, making the uncollapsed PDD a multiset. Now, let $\mathbb{A}$ be a multiset of the form $\mathbb{A} = \{a_i^{(j)} : i \in [1, \dots, n], j \in [1, \dots, n_i]\}$ where $n_i$ is the multiplicity of element $a_i$ and $a_i^{(j)}$ is the $j^{th}$ occurrence of element $a_i$. This multiset can be turned into a weighted set by assigning each element $a_i$ with the weight $\frac{n_i}{n}$. We can recover the influence of multiplicity by the use of weights in our model.

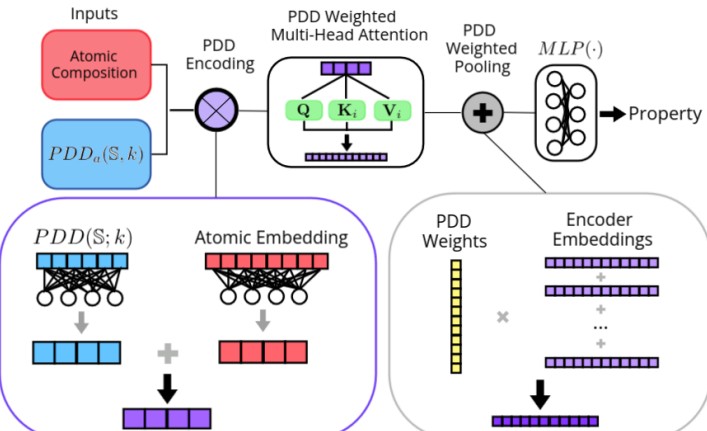

Figure 2: Overview of the architecture of the Periodic Set Transformer. PDD encoding is used to combine the structural information in the PDD with atomic embeddings containing compositional information. The weights of the PDD are incorporated in the attention mechanism and during the pooling of the embeddings to define the multiplicity of the input set.

When a periodic crystal has its unit cell modified, the proportion of each atom is expanded or reduced. The use of weights captures this behavior in the form of a concentration or frequency. In

this way, we can describe a potentially arbitrarily large crystal structure in terms of a distribution consisting of the atoms within a crystal that exhibit a particular behavior.

We use an attention mechanism to find the interactions between members of the set. The rows of the PDD contain the pairwise distance information, but they do not indicate which atoms these distances correspond to. Application of the attention mechanism can help the model learn these interactions.

Let $\boldsymbol{R} \in \mathbb{R}^{m \times k}$ be the PDD matrix containing $m$ rows without the associated weight column. Let $\boldsymbol{w} \in \mathbb{R}^{m \times 1}$ be the column vector containing the weights from the PDD matrix. The initial embedding is $\boldsymbol{X}^{(0)} = \boldsymbol{R}\boldsymbol{W}_d$ where $\boldsymbol{W}_d \in \mathbb{R}^{k \times d}$ is the initial trainable weight matrix. The embedding is updated according to:

$$\boldsymbol{X}^{(1)} = \boldsymbol{X}^{(0)} + SLP\left(\sigma\left(\frac{\boldsymbol{Q}\boldsymbol{K}^T}{\sqrt{d}}\right)\boldsymbol{V}\right) \tag{1}$$

where $\boldsymbol{Q} = \boldsymbol{X}^{(0)}\boldsymbol{W}_Q$, $\boldsymbol{K} = \boldsymbol{X}_i^{(0)}\boldsymbol{W}_K$ and $\boldsymbol{V} = \boldsymbol{X}^{(0)}\boldsymbol{W}_V$ and $d$ is the embedding dimension of the weight matrices for the query, key, and value $\boldsymbol{W}_Q, \boldsymbol{W}_K$, and $\boldsymbol{W}_V$ respectively as described in Vaswani et al. (2017). The function $\sigma$ is the softmax function with the PDD weights integrated into it; $\sigma$ is applied to each row $\boldsymbol{z}$ of the input matrix, and $i$ and $j$ are used to index entries in $\boldsymbol{z}$ and $\boldsymbol{w}$. The $i^{th}$ entry of the output vector is defined by:

$$\sigma(\boldsymbol{z})_i = \frac{w_i e^{z_i}}{\sum_{j=1}^m w_j e^{z_j}} \tag{2}$$

The result is passed through a single-layer perceptron $SLP$. The layer normalization order described by Xiong et al. (2020) is used for increased stability during training. This process is repeated $l$ times; this determines the depth of the model. The embeddings are finally pooled into a single vector by reincorporating the PDD weights into a weighted sum of the row vectors $\boldsymbol{x}_i$ of the final embedding $\boldsymbol{X}^{(l)}$.

$$\boldsymbol{x} = \sum_i w_i \boldsymbol{x}_i \tag{3}$$

This final embedding can be passed to a perceptron layer to predict the property value.

This is very similar to dot product attention as defined by the original Transformer article (Vaswani et al., 2017), but it is a modified version of self-attention that incorporates the PDD weights, though it is not limited to this. This version of self-attention can be applied to a weighted set or distribution. An overview of the Periodic Set Transformer (PST) architecture is shown in Figure 2.

## 3.2 PDD ENCODING

While structure is a powerful indicator of lattice energy, there may be datasets in which it is not the primary differentiator of a set of crystals. In such cases, the composition of the atoms contained within the material has a heavy influence. The previously described transformer does a good job of utilizing the structural information within the PDD but does not provide an obvious way to include atomic composition. Here we will describe a method to incorporate this information while maintaining all benefits of the PDD.

Transformers for natural language processing tasks use positional encoding to allow the model to distinguish the position of words within a given sentence (Dufter et al., 2022). A recent transformer model, *Uni-Mol* (Zhou et al., 2023), which performed property prediction for molecules (among other tasks), used *3D spatial encoding* first proposed by Ying et al. (2021) to give the model an understanding of each atom's position in space, relative to one another. This encoding is done at the pair level, using the Euclidean distance between atoms and a pair-type aware Gaussian kernel (Shuaibi et al., 2021). A transformer model for finite $3D$ points clouds is provided by Zhao et al. (2021) via *vector* attention. The case for crystals is more difficult because they are not bounded in size and encoding at the pair level can result in discontinuity under perturbations. Fortunately, by using the rows of the PDD we can distinguish each atom with structural information. We refer to this as *PDD encoding*.

When rows are grouped together, they are done so by having the same $k$-nearest neighbor distances. Though rare, it is possible for rows corresponding to different atom types to be collapsed. If this occurs, the selection of either atom type will result in information loss. To prevent this, we add the condition that the groups must be formed on the basis of having the same $k$-nearest neighbor distances and the same atomic species. We will refer to this as the *atomistic PDD* and distinguish it with the notation $PDD_a(S; k)$.

For a periodic set $S$, let $PDD_a(S; k)$ be the resulting PDD matrix with parameter $k$. Let $\boldsymbol{R}$ be $PDD_a(S; k)$ without the initial weight column and $\boldsymbol{T}$ be the matrix whose rows correspond to the atomic embedding of the type of atom associated with each row of $PDD_a(S; k)$. The initial set of embeddings for the attention mechanism is defined as $\boldsymbol{X}^{(0)} = \boldsymbol{R}\boldsymbol{W}_s + \boldsymbol{T}\boldsymbol{W}_c$ where $\boldsymbol{W}_s$ and $\boldsymbol{W}_c$ are initial embedding weights. By starting with a linear embedding, the PDD row can be transformed to match the dimension of composition embedding. The parameter $k$ used can then be changed as needed to include distance information from further neighbors. As with positional encoding, we displace the atomic embedding by the PDD encoding vector by summing. The rest of the model continues as before.

# 4 PREDICTION OF LATTICE ENERGY

The first dataset to be considered contains simulated molecular crystals created by Pulido et al. (2017) during the crystal structure prediction using quasi-random sampling (Case et al., 2016). This data is subsetted by the underlying molecule, e.g. T2. During structure prediction, crystals are generated while traversing the potential energy surface and their lattice energy is calculated using *DMACRYS* (Price et al., 2010) to determine their stability.

Prediction of lattice energy is done in three different scenarios. In the first, the model will be applied to a single set of molecular crystals with the same composition using $80/10/10$ training, validation, and testing splits. Next, the model is applied to multiple sets of crystals each with different underlying molecules using the same splits. The final experiment consists of the application of the model to a set of crystals with an underlying molecule it has not seen in training. The seen data is split $90/10$ for training and validation. To make predictions the PST described in section 3.1 is used with only the PDD as input and no knowledge of the composition.

Invariants are usually used to discern crystals by measuring differences between their structure. Here, the goal is to demonstrate the effectiveness of using an invariant as a representation for a machine learning algorithm. Even when compositional information is not present, the PDD can distinguish crystals with the EMD from the changes in the pairwise distances that occur when the species of atoms are changed. Whether the same distinction can be made in the context of a learning algorithm has previously not been shown.

We make a comparison to another invariant that has been used to predict lattice energy. Average-minimum-distance (Widdowson et al., 2022) was used as input for a Gaussian regression model (Ropers et al., 2022). In Table 1 we list the performance on the test set of this AMD model to allow comparison between invariants.

Our model reduces the mean-absolute-error (MAE) rate by $21\%$ compared to the Gaussian Process Regression technique which utilizes AMD (Ropers et al., 2022). The mean-absolute percentage error (MAPE) is also reduced by $1.63\%$. While we use $k = 60$ nearest neighbors for constructing the PDD, the AMD model uses $k = 500$ to achieve its best results. Despite the use of this additional information, the model using the PDD still performs more accurately.

In the second row of Table 1 we list the results of the second experiment. The previous task is extended to a dataset of crystals that contains different underlying molecules. These molecules have different compositions. This compositional information is not contained within the PDD (and thus, not in our input). The model will have to discern crystals solely by their structure. While the overall MAE has increased slightly, the percentage error has decreased. The domain on which the lattice energies lie is different for each type of crystal. Using the PDD alone is enough for the algorithm to distinguish the crystal types and predict lattice energy accordingly.

The final experiment uses the data from the P1, P1M, and P2 crystals in the training and validation data. The test set consists of the P2M crystal, which is not part of the either training or validation

Table 1: Results of lattice energy prediction using the PDD with the PST and AMD with Gaussian regression on three different tasks using $80/10/10$ training, validation, and testing splits. **(a)** uses only experimentally generated structures of the T2 molecular crystal based on triptycene. **(b)** adds two additional sets of molecular crystals, P1 (based on pentiptycene) and S2 (based on spiro-biphenyl). **(c)** uses P1, P1M (methylated analogue of P1) and P2 (benzimidazolone series pentiptycene) in the training and validation set using a $90/10$ split. The test set consists only of P2M, the methylated analogue of P2.

|  | Train | Train Size | Test | Test Size | Invariant | MAE (kj/mol) ↓ | MAPE ↓ |
|---|---|---|---|---|---|---|---|
| **(a)** | T2 | 4,630 | T2 | 578 | AMD | 4.79 | 4.31% |
|  |  |  |  |  | PDD | 3.76 | 2.68% |
| **(b)** | T2, P1, S2 | 14,547 | T2, P1, S2 | 1,819 | AMD | 4.68 | 2.83% |
|  |  |  |  |  | PDD | 4.11 | 2.52% |
| **(c)** | P1,P1M,P2 | 22,995 | P2M | 7,352 | AMD | 12.99 | 6.89% |
|  |  |  |  |  | PDD | 7.24 | 3.89% |

set. This experiment is the closest to real-world conditions in which new crystals often arise and finding the stable forms is crucial, but information on their lattice energy is unavailable.

When lattice energy is calculated using ab initio calculations, the range of the energies varies from crystal to crystal. When introducing a new type of crystal for our algorithm to make predictions on, this can become a problem as extrapolation to unvisited parts of the lattice energy range can result in high error rates. Fortunately, lattice energies between different types of crystals, are not usually meant to be compared. Instead, they are generated for potential polymorphs of a crystal in an effort to find those with the highest stability for synthesis. We can make use of this fact when applying our model to novel crystals. It is not necessary to predict the correct range of lattice energies; instead, the model needs to be able to predict the lattice energies of the various structures such that their ordering according to their lattice energy is correct. This task could feasibly be turned into a *learning-to-rank* problem (Liu et al., 2009), but as a regression task, it allows for a more general approach since the predicted lattice energies can be ordered after the fact.

Each dataset has its lattice energies shifted by the mean lattice energy towards zero. By doing this they each maintain their distribution but now overlap around the origin. The model is trained and validated based on this shifted data. The MAE of the predictions on the test set after they have been shifted back is 7.24 kJ/mol and the MAPE is $3.89\%$.

While the MAE and MAPE are higher than in previous experiments, the improvement over AMD is more significant. The majority of errors come from underestimating the lattice energy. The datasets tend to grow sparser in these areas where lattice energy is lower as this is where the most stable structures tend to lie. Having a false positive (predicting a higher energy structure to be lower) increases the number of potentially stable structures. False negatives are more impactful as they may result in a structure not being considered entirely due to its seemingly low stability.

## 5 PREDICTION OF MATERIALS PROJECT PROPERTIES

The model will be applied to the data within the Materials Project. In order to make fair comparisons to other models we report the performance according to *Matbench* (Dunn et al., 2020), which contains data for various crystal properties. The error rates are reported using five-fold cross-validation with standardized training and testing sets for each fold. Further, tuning is done according to the models' authors and thus our model can be compared to others more fairly.

The crystals in the Materials Project are highly diverse in composition. For all predictions, we include the composition of the crystal with PDD encoding. The composition of each atom is embedded with a 92 dimensional vector with various one-hot encoded atomic properties. This is the exact same embedding used by CGCNN (Xie & Grossman, 2018).

### 5.1 EXPERIMENTAL RESULTS

In Table 2 we report the average MAE across the five test sets. We include the reported accuracies of other models to allow for comparison. The selection of models aims to present a high diversity in approaches. *CGCNN* (Xie & Grossman, 2018) is a prominent work in crystal property prediction.

The application of the graph convolutional layer developed in this work is still relevant and used in even more recent works (Choudhary & DeCost, 2021; Cao et al., 2023; Das et al., 2022). *CrabNet* (Wang et al., 2021) is the only other Transformer model listed on Matbench. This model, in terms of architecture, is the most similar to our own. *MEGNet* (Chen et al., 2019) is also a GNN that uses a different message-passing function compared to the distance-weighted convolution of CGCNN. Additionally, the edge embeddings of the graph take into account the atomic attributes of the nodes they are adjacent to. It also keeps track of a global state embedding that is updated along with the node and edge attributes. *coGN* is a GNN that includes angular and dihedral information through the use of line graphs. *Crystal Twins* (CT) (Magar et al., 2022) is a model based on CGCNN that uses self-supervised learning to create embeddings based on maximizing the similarity between augmented instances of a crystal.

Table 2: Five-fold cross-validation prediction MAE for properties of the crystals in the Materials Project. Bold values indicate the best (lowest) error rate while underlined values indicate the second-best error rate. PST performance is reported using PDD Encoding with a tolerance of $10^{-4}$ and $k = 15$.

| Property | Units | PST | CGCNN | MEGNet | CrabNet | coGN | CT |
|---|---|---|---|---|---|---|---|
| Formation Energy | $eV/atom$ | 0.038 | 0.034 | 0.025 | 0.0862 | **0.021** | 0.037 |
| Band Gap Energy | $eV$ | 0.266 | 0.297 | 0.193 | 0.266 | **0.156** | 0.264 |
| Shear Modulus | $log_{10}(GPa)$ | 0.078 | 0.089 | 0.087 | 0.101 | **0.069** | 0.086 |
| Bulk Modulus | $log_{10}(GPa)$ | 0.056 | 0.071 | 0.067 | 0.076 | **0.053** | 0.067 |
| Refractive Index | n/a | **0.300** | 0.599 | 0.339 | 0.323 | 0.309 | 0.417 |
| Phonon Peak | $1/cm$ | 36.69 | 57.76 | **28.76** | 55.11 | 29.71 | 48.86 |
| Exfoliation Energy | $meV/atom$ | **36.61** | 49.24 | 54.17 | 45.61 | 37.16 | 46.69 |

From the mean MAE, the PST performs significantly better than CrabNet in all but band gap energy where our model produces a fractionally higher error rate. Further, the PST performs better than CGCNN on all properties except formation energy. While the architecture of PST and CGCNN are significantly different, the information used is the most comparable between all the models. Both models use the distances between the $k$-nearest neighbors of atoms within the unit cell and the same embedding for atomic properties. Despite this and the additional structural information provided by using edge embeddings in the graph representation, PST is still able to perform more accurately across all but one property. The PST also outperforms CT, a pre-trained model, in five of the seven properties with the band gap and formation energy MAE being slightly higher.

The root cause of the disparity in performance between the PST and other models on formation and band gap energy can be difficult to discern. MEGNet updates edge embeddings within the message-passing layer, while CGCNN and CT do not. All four take a graph approach to their representations but the performance of CGCNN and CT (which is based on CGCNN) are significantly worse. CoGN takes this a step further and updates the original edges with message-passing from the derived line graph, allowing the inclusion of angular information. The edges of the line graph are further updated by its line graph, incorporating dihedral angles. These updates likely result in a richer learned representation, which is seemingly necessary for these larger datasets. This points to a current limitation for the PST. By using PDD encoding, we effectively limit opportunities for such updates to a single embedding representing both the atom's properties and its structural behavior.

## 5.2 ABLATION STUDY

In Table 3 we list the results for each "component" within the Periodic Set Transformer. In the row indicating PDD as the component, we train and test the model using only the structural information within the PDD; this method is the same as in the previous lattice energy experiments. In the "Composition" component we pass the atomic encoding for the elements in the crystal and their concentration in the form of the PDD weights to the model without the PDD encoding.

By separating out each component of the model, we can interpret the importance of each to a particular property. Properties that experience a more significant decrease in performance when the PDD encoding is not used, can be ascribed to be more dependent on structural information. In all cases, the combination of both the composition and PDD encoding results in significantly lower error rates. We can conclude that this encoding method is effective in combining the structural and compositional information of a crystal structure.

| Property (units) | Component MAE ↓ | | |
|---|---|---|---|
| | Composition | PDD | PST |
| Band Gap $(eV)$ | 0.332 | 0.612 | **0.266** |
| Formation $(eV)/atom$ | 0.087 | 0.440 | **0.038** |
| Shear Modulus $log_{10}(GPa)$ | 0.111 | 0.140 | **0.078** |
| Bulk Modulus $log_{10}(GPa)$ | 0.083 | 0.122 | **0.056** |
| Refractive Index | 0.342 | 0.502 | **0.300** |
| Phonon Peak $1/cm$ | 58.47 | 87.29 | **36.69** |
| Exfoliation $meV/atom$ | 49.22 | 40.70 | **36.61** |

Table 3: Effect of PDD encoding on prediction MAE of the Materials Project crystals. Results are separated by input components where "Composition" uses only the atomic embeddings detailed in appendix Table 6 and "PDD" uses only the PDD. Errors in bold indicate the best performance and underlined errors indicate second-best performance (lower is better ↓).

In Table 4 the effect of including the PDD weight in the attention mechanism described in Equation 2 and in the pooling layer described by Equation 3 is listed. As expected, the exclusion of weights from both the attention mechanism and pooling decreases accuracy significantly. Doing this removes all indications of multiplicity making discernment of crystals more difficult. The inclusion of the weights in the pooling layer is more impactful than when applied in the attention mechanism. The use of the weights in the pooling layer alone allows the model to perform better when the number of samples in the dataset is low. Datasets with fewer samples likely have less diversity amongst their crystals, making the need for recognizing the multiplicity of atoms less necessary.

| Property (units) | PDD Weight Inclusion MAE ↓ | | | |
|---|---|---|---|---|
| | No Weights | Attention Only | Pooling Only | PST |
| Band Gap $(eV)$ | 0.298 | 0.279 | **0.263** | 0.266 |
| Formation $(eV)/atom$ | 0.063 | 0.049 | 0.042 | **0.038** |
| Shear Modulus $log_{10}(GPa)$ | 0.080 | 0.079 | 0.078 | **0.078** |
| Bulk Modulus $log_{10}(GPa)$ | 0.061 | 0.057 | 0.057 | **0.056** |
| Refractive Index | 0.306 | 0.303 | 0.300 | **0.300** |
| Phonon Peak $1/cm$ | 35.22 | 35.15 | **33.49** | 36.69 |
| Exfoliation $meV/atom$ | 37.45 | 38.52 | **34.65** | 36.61 |

Table 4: Effect of including the PDD weights as defined by Equations 2 and 3 on prediction MAE of the Materials Project crystals. Results for "No weights" use mean pooling and a normal softmax function. Errors in bold indicate the best performance and underlined errors indicate second-best performance (lower is better ↓).

## 6 CONCLUSION

The PDD is a representation for periodic crystals which is invariant to rigid motion and independent of unit cell. By using weights and creating a distribution, the PDD is able to represent an infinitely spanning object by its finite forms of behavior. Further, by collapsing rows in the PDD, the resulting representation can also be much smaller in comparison to the number of atoms within the unit cell, even when the cell is reduced.

We have provided a way to use the PDD in an attention-based model to predict lattice energies using a modified version of self-attention and PDD-weighted readout. The robustness of this model is shown by applying it to multiple crystals with different underlying molecules. Additionally, we test the model on a crystal with a novel underlying molecule that has not yet been seen in the training or validation data. The model is extended to include atomic composition by using PDD encoding, a form of positional encoding for distinguishing structural differences in crystals. This model, the Periodic Set Transformer, is capable of producing results on par or even exceeding much more commonly used architectures for crystal property prediction like GNNs in several properties. Further, our transformer model outperforms a previously developed transformer for crystal property prediction in six out of the seven properties.

## 7 REPRODUCIBILITY

The associated source code is available in the supplementary material. The code contains what is necessary to re-run the experiments done in sections 4 and 5.1. It also contains the source code necessary to recreate Figures 4a and 4b. Details of how these plots are created are included in section B. The individual predictions for the Materials Project data are contained in *JSON* format within the repository. Proofs for the properties of the PDD mentioned in (Widdowson & Kurlin, 2022, Problem 1.1) are included in the original paper Widdowson & Kurlin (2022). More details on the actual implementation of the model, data pre-processing, and training are contained in section G. The dataset for the crystals used in the lattice energy experiments is available at https://eprints.soton.ac.uk/404749/. The data from the Materials Project is automatically downloaded through the code in the supplementary material.

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

## A  EQUIVALENCE OF DISTRIBUTIONS IN THE PST

If a PDD is arbitrarily expanded or collapsed, the PST should produce the same results as these PDDs are considered equivalent. The same can be said of any input distribution. This is proven here:

**Lemma A.1.** *Let $\mathbb{A}$ and $\mathbb{B}$ be weighted multisets each containing elements from the set $\mathbb{S} = \{\boldsymbol{x}_1, \dots \boldsymbol{x}_n\}$. Each element $\boldsymbol{x}_i \in \mathbb{R}^{1 \times n}$ occurs with multiplicity $m_i^{(a)} \in \mathbb{N}^+$ and $m_i^{(b)} \in \mathbb{N}^+$ in $\mathbb{A}$ and $\mathbb{B}$ respectively. Each element also carries weight $w_i^{(a)} \in \mathbb{R}^+$ and $w_i^{(b)} \in \mathbb{R}^+$ for $\mathbb{A}$ and $\mathbb{B}$ respectively. The application of the Periodic Set Transformer will yield equivalent output if*

$$w_i^{(a)} m_i^{(a)} = w_i^{(b)} m_i^{(b)}, \quad \forall i \in \{1, \dots, n\} \tag{4}$$

*Proof.* To prove that the output of the PST is equivalent for $\mathbb{A}$ and $\mathbb{B}$, it is sufficient to prove that the output of $\sigma$ (defined in Equation 2) and the pooling layer (defined in Equation 3) are the same for $\mathbb{A}$ and $\mathbb{B}$. Let $\boldsymbol{q}_i = \boldsymbol{x}_i \boldsymbol{W}_Q$, $\boldsymbol{k}_i = \boldsymbol{x}_i \boldsymbol{W}_K$, and $\boldsymbol{v}_i = \boldsymbol{x}_i \boldsymbol{W}_V$ be the query, key, and value vectors produced by the weight matrices $\boldsymbol{W}_Q, \boldsymbol{W}_K, \boldsymbol{W}_V \in \mathbb{R}^{n \times d}$. First, note the pre-softmax attention weight from $\boldsymbol{x}_i$ to $\boldsymbol{x}_j$ can expressed as:

$$a_{ij} = \frac{\boldsymbol{q}_i \boldsymbol{k}_j^T}{\sqrt{d}} \tag{5}$$

and is independent of both weight and multiplicity. Further, notice that the summation of an expression over $\mathbb{A}$ or $\mathbb{B}$ can be rewritten using its multiplicities. For $\mathbb{A}$ this is done like so:

$$\sum_{t=1}^{|\mathbb{A}|} (\cdot) = \sum_{s=1}^{n} m_s^{(a)} (\cdot)$$

Using this equivalence, the function $\sigma$, used to calculate the attention weight from $\boldsymbol{x}_i$ to $\boldsymbol{x}_j$ for $\mathbb{A}$ and $\mathbb{B}$, is written as:

$$\alpha_{ij}^{(a)} = \frac{w_i^{(a)} exp(a_{ij})}{\sum_{k=1}^{n} w_k^{(a)} m_k^{(a)} exp(a_{ik})} \tag{6}$$

and

$$\alpha_{ij}^{(b)} = \frac{w_i^{(b)} exp(a_{ij})}{\sum_{k=1}^{n} w_k^{(b)} m_k^{(b)} exp(a_{ik})} \tag{7}$$

Using the condition provided by Equation 4, Equation 7 can be rewritten as:

$$\alpha_{ij}^{(b)} = \frac{w_i^{(b)} exp(a_{ij})}{\sum_{k=1}^{n} w_k^{(a)} m_k^{(a)} exp(a_{ik})} \tag{8}$$

$$\sum_{k=1}^{n} w_k^{(a)} m_k^{(a)} exp(a_{ik}) = \frac{w_i^{(b)} exp(a_{ij})}{\alpha_{ij}^{(b)}} \tag{9}$$

Substituting back into Equation 6 gives us:

$$\alpha_{ij}^{(a)} = \alpha_{ij}^{(b)} \frac{w_i^{(a)} exp(a_{ij})}{w_i^{(b)} exp(a_{ij})} \tag{10}$$

$$\alpha_{ij}^{(a)} = \alpha_{ij}^{(b)} \frac{w_i^{(a)}}{w_i^{(b)}} \tag{11}$$

The attention weights produced from $\mathbb{A}$ can now be expressed in terms of the attention weights produced from $\mathbb{B}$. The $j^{th}$ entry in the attention vector for $\boldsymbol{x}_i$ in $\mathbb{A}$ is:

$$y_{ij}^{(a)} = \sum_{j=1}^{n} \alpha_{ij}^{(a)} m_j^{(a)} v_{ji} \tag{12}$$

$$= \sum_{j=1}^{n} \alpha_{ij}^{(b)} \frac{w_i^{(a)}}{w_i^{(b)}} m_j^{(a)} v_{ji} \tag{13}$$

$$= \sum_{j=1}^{n} \alpha_{ij}^{(b)} \frac{w_i^{(a)}}{w_i^{(b)}} \left( \frac{w_i^{(b)} m_i^{(b)}}{w_i^{(a)}} \right) v_{ji} \tag{14}$$

$$= \sum_{j=1}^{n} \alpha_{ij}^{(b)} m_i^{(b)} v_{ji} \tag{15}$$

$$= y_{ij}^{(b)} \tag{16}$$

Thus, the resulting embeddings from the attention mechanism are equivalent. While the embeddings themselves are equivalent, the cardinality for each embedding still differs according to each multi-sets' multiplicity. The pooling described by Equation 3 fixes this. Let $\boldsymbol{z}_i$ be the final embedding for $\boldsymbol{x}_i$. The output vector $\boldsymbol{z}$ from the pooling for $\mathbb{A}$ is the sum:

$$\boldsymbol{z}^{(a)} = \sum_{i=1}^{n} w_i^{(a)} m_i^{(a)} \boldsymbol{z}_i \tag{17}$$

Substituting Equation 4 again we get,

$$\boldsymbol{z}^{(a)} = \sum_{i=1}^{n} w_i^{(b)} m_i^{(b)} \boldsymbol{z}_i \tag{18}$$

$$= \boldsymbol{z}^{(b)} \tag{19}$$

Thus, the output of the PST for both $\mathbb{A}$ and $\mathbb{B}$ are equivalent. $\qquad\square$

## B  APPLICATIONS OF THE POINTWISE DISTANCE DISTRIBUTION

In definition 3.2 we provided a formal definition for the PDD. Here, we will begin by providing an example of its construction.

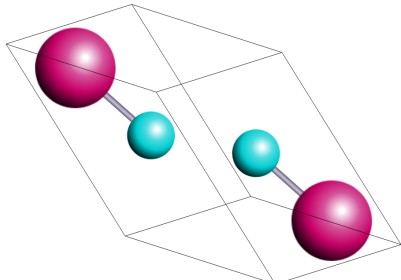

Figure 3: The unit cell of Lutetium-Silicon; Silicon is colored in teal and Lutetium in magenta.

Consider Lutetium-Silicon which is shown in Figure 3. The unit cell pictured contains a total of four atoms, 2 of which are Lutetium and 2 of which are Silicon. If we use $k = 2$ nearest neighbors, the PDD matrix of this periodic set $S$ before rows are grouped is

$$PDD(S; k) = \begin{pmatrix} 0.25 & 2.481 & 2.481 \\ 0.25 & 2.481 & 2.481 \\ 0.25 & 2.881 & 2.881 \\ 0.25 & 2.881 & 2.881 \end{pmatrix}$$

where the first column contains the weights for each row (atom). The second is the distance to the nearest neighbor and the third, the distance to the second nearest neighbor. The first two rows are identical, as are the final two. Because of this, each of the two groups of rows can be grouped into a single row as so,

$$PDD(S;k) = \begin{pmatrix} 0.5 & 2.481 & 2.481 \\ 0.5 & 2.881 & 2.881 \end{pmatrix}$$

The rows are already lexicographically ordered so this is the finalized PDD. If a different unit cell is selected, this collapsing of matrix rows will continue to yield the same result as the proportion of each atom will not change.

The PDDs of two periodic crystals can be compared using the Earth Mover's Distance (Rubner et al., 2000). The weights of each PDD can be considered the distribution we are comparing in the minimum flow problem. We can visualize a set of crystals by projecting these distances onto a two or three-dimensional plane.

Subfigure 4a shows the projection of the pairwise distances of the PDDs of one hundred samples from each of the three crystals in this experiment using Multi-Dimensional Scaling (MDS). This technique attempts to map a set of pairwise distances to specific $n$-dimensional space by minimizing the difference between the actual pairwise distances and the distances between the points in the projected space. Even with this error, we are able to establish three distinct clusters in two-dimensional space according to their respective crystal type.

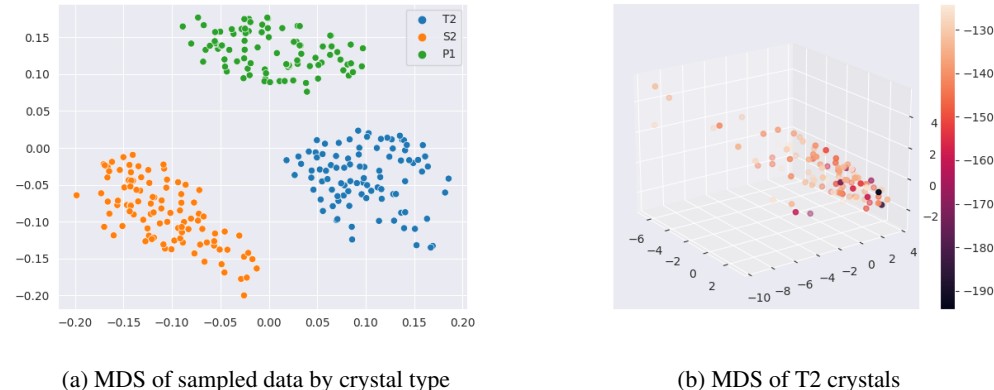

(a) MDS of sampled data by crystal type       (b) MDS of T2 crystals

Figure 4: Multi-dimensional scaling projection on to $\mathbb{R}^2$ and $\mathbb{R}^3$ for the pairwise distances of crystals between each other. Subfigure (a) projects three types of crystals using distances created by using the Earth Mover's Distance between PDDs for $k = 10$. Subfigure (b) is created using the MDS projection of pairwise distances between PDDs at $k = 100$ for one hundred random samples from the T2 crystals colored by lattice energy measured in kJ/mol.

Subfigure 4b shows the projection of the pairwise distances between the crystals' PDD onto three-dimensional space for the T2 dataset. The colors of the points in each plot signify the lattice energy for the crystal. The distinction here is not as pronounced; this is to be expected as the sampled crystals are much more similar in their structure and thus the distances between the PDDs are smaller. Nonetheless, there is a discernible trend, and crystals that congregate near each other do share similar lattice energies.

## C  ADDITIONAL DETAILS ON LATTICE ENERGY RESULTS

The histogram in Figure 5a shows the lattice energies (in kJ/mol) of the three crystals within the dataset. Figure 5b shows the comparison of the predictions of the model against the ground truth values.

Predictions that have lower errors will have their point placed closer to the line colored in blue. The bulk of the points share their error both below and above the true lattice energy as we would expect

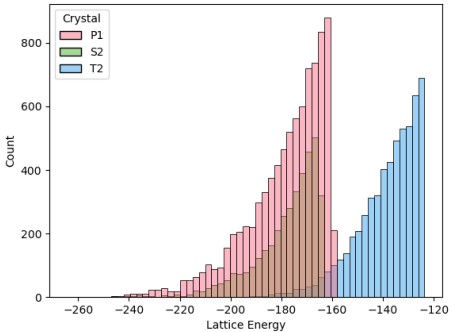 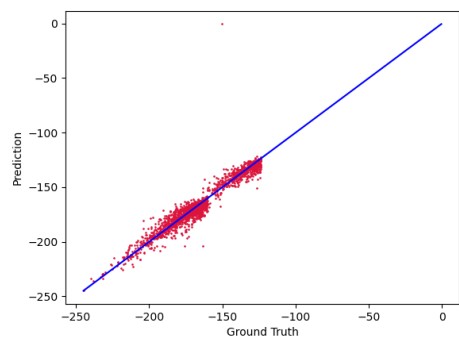

(a) Distribution of ground truth lattice energies.  (b) Ground truth lattice energy vs. predictions

Figure 5: (a) The distribution of the lattice energies of the T2, S2 and P1 crystals. (b) The predictions of T2, S2, and P1 compared to the ground truth lattice energies in kJ/mol.

in a model without bias. There are a few outliers, in particular, a single crystal from the T2 dataset has a predicted lattice energy of just $-0.41$. Prevention of such errors can be mitigated by using a different loss function than MAE. In particular, mean-squared error (MSE) and Huber loss can hedge against outlying errors by increasing their contribution to the loss function. We choose to not present the results using these loss functions as MAE still provides better results for MAE and MAPE.

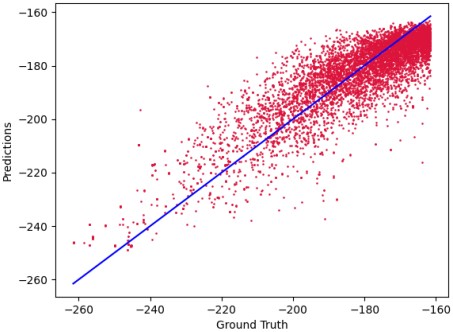 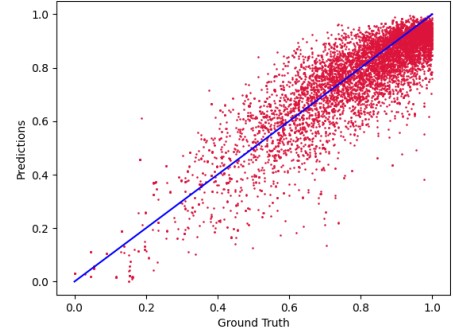

(a) Ground Truth vs. Predictions re-scaled according to mean.  (b) Ground Truth vs. Predictions normalized between zero and one.

Figure 6: (a) Comparison of predictions vs. ground truth of the test set after the predictions are scaled back by the mean of the lattice energies. (b) The comparison of predictions vs. ground truth after both have been normalized between zero and one.

In Figure 6a we plot the original predictions on P2M after they have been re-scaled back away from the mean lattice energy. In Figure 6b, we re-scale the prediction and the ground truth values to between zero and one. By doing this, we can see the ordering of the predictions compared to the true lattice energies. The scatter plot in Figure 6b allows us to compare lattice energies relative to other predictions.

If the error rate produced by the final experiment of section 4 is inadequate, we can supplement the training with predictions from classical methods. For a novel crystal, ab initio calculations can be used to generate lattice energies for a small subset of structures. These samples can be integrated into the training set to improve the overall results. In order to make this practical, we can only produce lattice energies for a limited portion of structures; we limit this to $10\%$ (or under) of the total structures.

Table 5: Effect of including portions of the P2M data into the training set on prediction accuracy.

| Samples | Percent of P2M Data | Test MAE (kj/mol) ↓ | Test MAPE ↓ |
|---------|--------------------|--------------------|-------------|
| 0 | 0% | 7.24 | 3.89% |
| 367 | 5% | 5.82 | 3.17% |
| 735 | 10% | 5.50 | 3.02% |

Table 5 displays the result of adding this supplemental data to the training set. As expected, the addition of data decreases MAE and MAPE. Even with as few as 367 samples (or $5\%$ of the total dataset), the reduction in error is significant. This process experiences diminishing returns as the amount of the original dataset is used in the training set. Though the MAE and MAPE continue to decrease, it is questionable whether or not spending time generating the instances using classical methods is worth the additional performance gains.

## D    ATOMIC EMBEDDINGS

To represent atoms we use the same 92 dimensional embedding used in Xie & Grossman (2018). The composition of each atom is embedded according to nine properties with respective dimensions listed in Table 6 in a one-hot encoding fashion. If the atomic property is continuous, the one-hot encoding is done with respect to bucketed values which are spread evenly across dimensions.

Table 6: Atom feature embeddings by atomic property.

| Property | Dimensionality |
|----------|----------------|
| Group Number | 18 |
| Period | 9 |
| Electronegativity | 10 |
| Covalent Radius | 10 |
| Valence Electrons | 12 |
| First Ionization Energy | 10 |
| Electron Affinity | 10 |
| Block | 4 |
| Atomic Volume | 10 |

## E    ADDITIONAL DETAILS ON MATERIALS PROJECT RESULTS

### E.1    EFFECT OF $k$-NEAREST NEIGHBORS

The PDD encoding can be said to be parameterized by two values, the collapse tolerance and the number of $k$-nearest neighbors. The integer $k$ determines the dimensionality of initial PDD encoding embedding. As $k$ increases, it retains all information from the previous values of $k$. The initial nearest neighbor distances are the most important and the embedding has diminishing returns after this. For any value of $k > 1$, the PDD is invariant. Thus, if the PDD is different, the crystals are guaranteed to be structurally different. In order for the PDD to be distinct such that if any two crystals are different, their PDDs are different, we need the property of generic completeness. This property is given provided the lattice $\mathbb{L}$ and sufficiently large $k$. An upper bound on this $k$ is when all distances in the last column of the PDD are larger than twice the covering radius of the lattice $\mathbb{L}$ of the periodic set. We would expect the performance of the model to increase up until this point.

The upper bound on $k$ can be exceedingly large so we implement a heuristic to find a lower bound that is more computationally efficient. As $k$ increases, the number of rows collapsed in the PDD will either stay the same or decrease. The lower bound on $k$ can be considered an integer large enough that the groups established at the upper bound of $k$ are the same as this lower bound. Each crystal could have a different $k$ for which this requirement is met. Our encoding method prohibits the use of a dynamic $k$ value, therefore we need a consistent value for $k$ that can be applied to all crystals in the dataset. The results in Table 2 use $k = 15$. At this value of $k$ the previously mentioned lower bound is satisfied for $99.1\%$ within the formation energy dataset. Increases in $k$ past this cause marginal

improvements to this coverage that were deemed insufficient when the increased computational cost is considered.

| Property (units) | MAE by $k$-Nearest Neighbors PDD Encoding ↓ | | | |
|---|---|---|---|---|
| | 5 | 10 | 15 | 20 |
| Band Gap   $(eV)$ | 0.275 | 0.268 | **0.266** | 0.267 |
| Formation   $(eV)/atom$ | 0.042 | 0.039 | 0.038 | **0.038** |
| Shear Modulus   $log_{10}(GPa)$ | 0.088 | 0.080 | **0.078** | 0.079 |
| Bulk Modulus   $log_{10}(GPa)$ | 0.061 | 0.058 | 0.056 | **0.055** |
| Refractive Index | 0.318 | 0.309 | 0.300 | **0.299** |
| Phonon Peak   $1/cm$ | 38.10 | 36.98 | 36.69 | **35.82** |
| Exfoliation   $meV/atom$ | 45.77 | 40.70 | 36.61 | **35.11** |

Table 7: Prediction MAE on the Materials Project crystals for various $k$-nearest neighbors PDD Encoding at a collapse tolerance of $10^{-4}$. Errors in bold indicate the value of $k$ with the best performance and underlined errors indicate the second-best performance (lower is better ↓).

The error rates listed in Table 7 vary the value of $k$ and report the resulting mean MAE across the five folds. As expected, the lower values of $k$ result in higher MAE. Increasing $k$ eventually causes the error rates to stop decreasing. This is also in line with what would be expected as the PDD has enough information to distinguish itself and additional distances are unnecessary.

### E.2   EFFECT OF COLLAPSE TOLERANCE

The collapse tolerance dictates which rows of the PDD will be collapsed. As this parameter increases, the size of the grouped rows will increase. Once rows are grouped, their distances are averaged in the row which represents the group. The change in this averaged row is proportional to the size of the collapse tolerance. In $PDD(S; k)$, as the collapse tolerance approaches infinity, the PDD will decrease in the number of rows until it consists of just a single row with a weight equal to one. In $PDD_a(S; k)$, the same increase in collapse tolerance will result in a number of rows within the PDD equal to the number of unique elements within the crystal. In both cases, a collapse tolerance that is large enough will result in information loss, eventually increasing errors in predictions.

The collapse tolerance is varied and then applied to the Materials Project crystals. The results of this experiment are listed in Table 8.

| Property (units) | MAE by Collapse Tolerance in PDD Encoding ↓ | | | |
|---|---|---|---|---|
| | 1.0 | $10^{-2}$ | $10^{-4}$ | 0.0 |
| Band Gap   $(eV)$ | 0.281 | 0.272 | 0.266 | **0.266** |
| Formation   $(eV)/atom$ | 0.045 | 0.042 | 0.038 | **0.037** |
| Shear Modulus   $log_{10}(GPa)$ | 0.078 | 0.079 | **0.078** | 0.078 |
| Bulk Modulus   $log_{10}(GPa)$ | 0.057 | 0.057 | **0.056** | 0.056 |
| Refractive Index | **0.299** | 0.301 | 0.300 | 0.300 |
| Phonon Peak   $1/cm$ | 34.23 | 36.21 | 36.69 | **33.96** |
| Exfoliation   $meV/atom$ | 38.34 | 37.64 | **36.61** | 38.38 |

Table 8: Prediction MAE on the Materials Project crystals using PDD Encoding at various collapse tolerances with $k = 15$. Errors in bold indicate the collapse tolerance with the best performance and underlined errors indicate the second-best performance (lower is better ↓).

By increasing the collapse tolerance and reducing the number of rows within the PDD, we can increase the speed of computations. Thus, it is ideal to choose a tolerance that is maximal, while not sacrificing accuracy. The impact of the collapse tolerance on the size of representation is listed in Table 9. These values are calculated by dividing the number of rows in the PDD by the number of atoms in the unit cell. The number of atoms in the unit cell is used to determine the number of vertices in the crystals graph (Xie & Grossman, 2018; Choudhary & DeCost, 2021; Das et al., 2022). In this way, the size of our representation can be compared to that of popular graph-based models. Data for crystals typically come in the form of Crystallographic Information Files (CIF).

These files also indicate the amount of potential measurement error for the atomic positions. This is used as a guide and a collapse tolerance of $10^{-4}$ is used in the experiments for the results in Table 2. Sometimes, however, a higher collapse tolerance can act as a regularization technique that is useful on smaller datasets. This effect is seen in the results for refractive index and phonon peak. Overall, the collapse tolerance does not have a very large impact due to the prevention of rows corresponding to different atoms from being collapsed in the atomistic PDD.

| Property | Mean $|\mathbb{M}|$ | Size of Input | | | | Percentage of $|\mathbb{M}|$ | | | |
|---|---|---|---|---|---|---|---|---|---|
| | | 0.0 | $10^{-4}$ | $10^{-2}$ | 1.0 | 0.0 | $10^{-4}$ | $10^{-2}$ | 1.0 |
| Phonon Peak | 7.5 | 7.3 | 3.6 | 3.5 | 3.4 | 96.8% | 55.7% | 54.5% | 53.8% |
| Ref. Index | 16.9 | 16.5 | 6.7 | 6.2 | 5.8 | 97.7% | 49.1% | 46.3% | 44.1% |
| Bulk Modulus | 8.6 | 8.3 | 4.1 | 3.9 | 3.7 | 96.7% | 62.1% | 60.4% | 59.4% |
| Shear Modulus | 8.6 | 8.3 | 4.1 | 3.9 | 3.7 | 96.7% | 62.1% | 60.4% | 59.4% |
| Band Gap | 30.0 | 29.2 | 13.1 | 12.0 | 10.1 | 97.0% | 52.0% | 48.5% | 44.2% |
| Formation | 29.1 | 28.4 | 12.7 | 11.6 | 9.9 | 96.9% | 53.1% | 49.7% | 45.7% |
| Exfoliation | 7.2 | 7.1 | 3.6 | 3.3 | 3.2 | 98.8% | 55.9% | 51.9% | 51.5% |

Table 9: Size of the input representation for each dataset in the Materials Project at various collapse tolerances at $k = 15$ compared to the number of atoms in the unit cell $|\mathbb{M}|$. The size of the input refers to the cardinality of the input set determined by the number of rows in the atomistic PDD. The percentage of $|\mathbb{M}|$ is the input set's cardinality divided by the number of atoms in the unit cell, expressed as a percentage.

### E.3 PREDICTIONS ON MATERIAL PROJECT PROPERTIES

In Figure 7, the individual predictions for each of the five test sets (for each of the folds within the cross-validation) can be seen plotted against their true property values.

For all properties, areas along the domain that are less densely populated produce less accurate predictions. This is typical for most machine learning models as there are fewer samples there, and thus optimization of the loss function does not prioritize these data points. For the refractive index, this is particularly true as there are a few samples that have very high property values, but the model predicts them to lie much closer to where the main cluster of points is. Bulk and shear modulus both take on discrete values. The distance between these values decreases as the property's value goes up. In the scatter plot, this can be seen as the clusters of points eventually become so close that they appear to take on continuous values above $\approx 1.5 \ log_{10}(GPa)$. Consequently, this upper range of bulk and shear modulus values is where the model produces the most accurate results. The spread of the formation energy data is much more consistent than the other properties. This is further aided by the large sample size. Thus, the predictions are consistent along the domain of the energy values.

### E.4 DETAILED LOSS METRICS

The detailed loss metrics for our model are listed within Tables 10, 11, 12, 13, 14, and 15, 16. They contain the error rates using MAE, MAPE, and RMSE, as well as their standard deviations.

Table 10: Performance of the Periodic Set Transformer on prediction of shear modulus across each fold for various error measures.

| Measure | Mean | Min | Max | Std. |
|---|---|---|---|---|
| MAE | 0.0781 | 0.0771 | 0.0794 | 0.0008 |
| RMSE | 0.1193 | 0.1171 | 0.1222 | 0.0021 |
| MAPE | 0.0610 | 0.0592 | 0.0625 | 0.0013 |
| Max error | 0.9584 | 0.7577 | 1.139 | 0.1286 |

## F COMPUTATIONAL COST

The Transformer architecture allows for each member of the input set to be processed in parallel. GPU acceleration allows the model to be quite performant. In the final experiment for lattice energy,

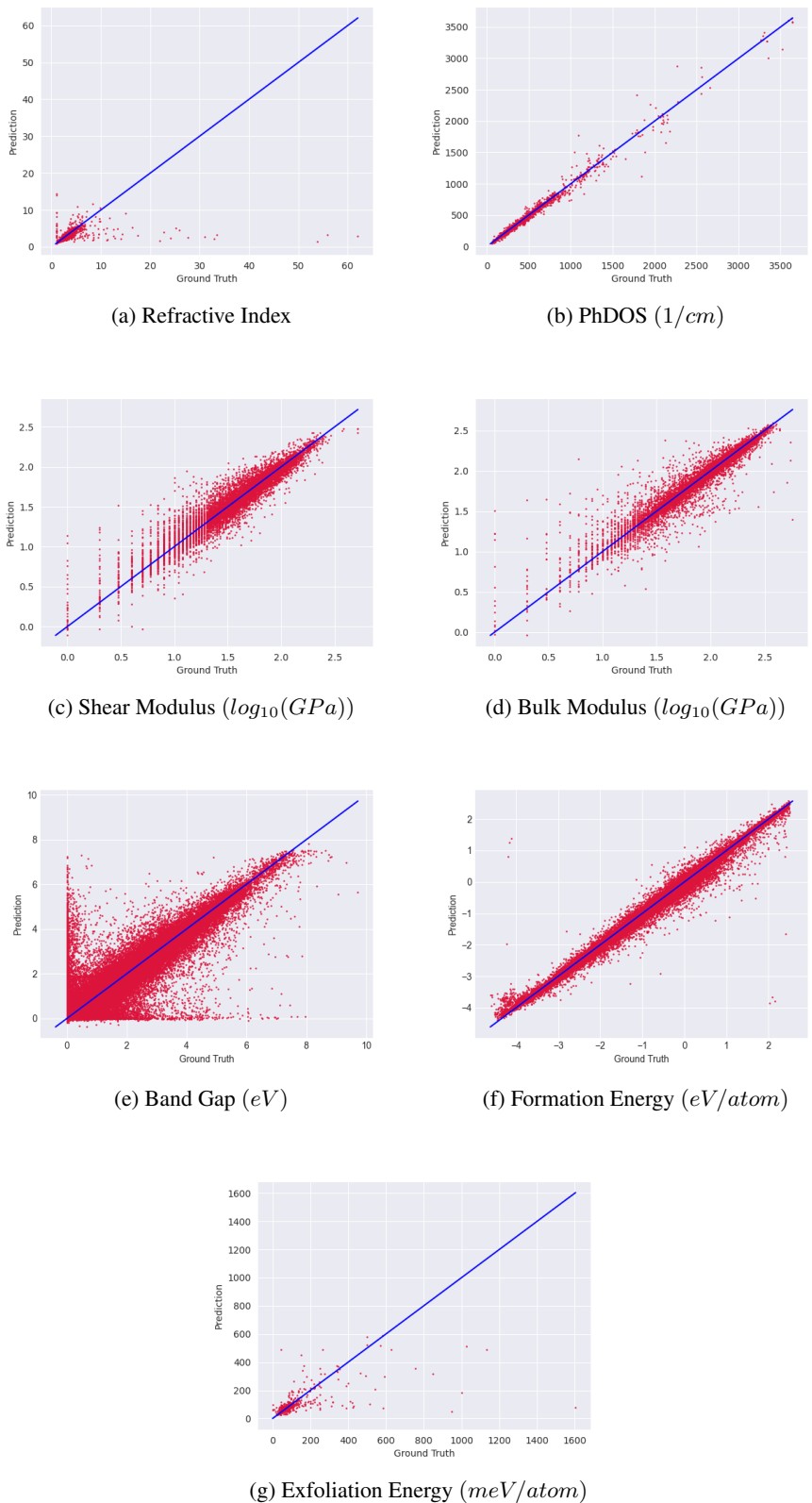

(a) Refractive Index

(b) PhDOS $(1/cm)$

(c) Shear Modulus $(log_{10}(GPa))$

(d) Bulk Modulus $(log_{10}(GPa))$

(e) Band Gap $(eV)$

(f) Formation Energy $(eV/atom)$

(g) Exfoliation Energy $(meV/atom)$

Figure 7: Comparison of predictions vs. ground truth across all folds for all seven Materials Project properties tested.

Table 11: Performance of the Periodic Set Transformer on prediction of bulk modulus across each fold for various error measures.

| Measure | Mean | Min | Max | Std. |
|---|---|---|---|---|
| MAE | 0.0565 | 0.0521 | 0.0600 | 0.0026 |
| RMSE | 0.1101 | 0.0986 | 0.1180 | 0.0074 |
| MAPE | 0.0378 | 0.0342 | 0.0430 | 0.0029 |
| Max error | 1.321 | 1.1690 | 1.5086 | 0.1179 |

Table 12: Performance of the Periodic Set Transformer on prediction of refractive index (dielectric) across each fold for various error measures.

| Measure | Mean | Min | Max | Std. |
|---|---|---|---|---|
| MAE | 0.3002 | 0.1844 | 0.4283 | 0.0799 |
| RMSE | 1.7137 | 0.6336 | 2.9488 | 0.8393 |
| MAPE | 0.0802 | 0.0641 | 0.0930 | 0.0121 |
| Max error | 32.720 | 14.511 | 59.017 | 18.313 |

we made predictions for all crystals in the P2M dataset (which consisted of $7,352$ structures) in $6.93$ seconds. This process took $6.93$ seconds for an average prediction rate of $1,061$ crystals per second or just under a millisecond per crystal on a Nvidia©RTX 3090. These crystals are particularly large in terms of the number of atoms within the unit cell when compared to those in the Materials Project. As such, this can be considered on the low end of performance with the potential number of predictions per second to be significantly larger.

In Table 17 we list the computation time for training each of the properties in the Materials Project. We also include the batch size and the number of samples for each dataset. Naturally, datasets with fewer samples take a shorter amount of time per epoch (and overall). The batch size is selected based on the number of samples with the primary focus of preventing overfitting. Because of this, the compute times could be reduced, but at the cost of potentially lower accuracy.

The times reported use the hyper-parameters of the accuracies reported in 2. The collapse tolerance used is $10^4$. Increasing the size of the tolerances results in more grouping to occur when creating the PDD. If this is done, the compute times can be reduced but potentially at the cost of accuracy (albeit a small cost).

## G  IMPLEMENTATION DETAILS

The Periodic Set Transformer is implemented using *PyTorch* (Paszke et al., 2019). There is also a version implemented using *Tensorflow* (Abadi et al., 2016), however, we have found this version to significantly underperform when compared to the PyTorch version. We believe this to be due to how the output of individual attention head output is handled in their respective implementations of Multi-head Attention.

Data pre-processing is fairly minimal. Each crystal comes in the form of a CIF. The CIF is read and converted into a `PeriodicSet` object. This functionality is provided by the *AMD* package (Widdowson et al., 2022). The PDD of each of the `PeriodicSet` objects is then calculated with the desired collapse tolerance and $k$ value. Each column is then normalized to between zero and one. This is not necessary for achieving the desired accuracy but it does significantly improve the speed of training by requiring fewer epochs.

With respect to the results in Table 2, training is done on each property with slightly different hyper-parameters. The hyper-parameters that govern PDD encoding, however, remain the same for all properties: a tolerance of $10^{-4}$ and $k = 15$. The dimensionality of the layers in the model, $d$, and the depth (number of encoders) were changed depending on the size of the dataset and its ability to reduce training error to an adequately small amount. If training error stopped decreasing too early, or plateaued at a high error rate, one (or both) of these hyper-parameters were increased. Similarly, weight decay (Krogh & Hertz, 1991) was added to datasets that contained fewer samples to aid in regularization. Typically the model that performs best on the validation set is chosen to

Table 13: Performance of the Periodic Set Transformer on prediction of formation energy across each fold for various error measures.

| Measure | Mean | Min | Max | Std. |
|---|---|---|---|---|
| MAE | 0.0387 | 0.0381 | 0.0392 | 0.0004 |
| RMSE | 0.0909 | 0.0800 | 0.1022 | 0.0093 |
| MAPE | 0.3044 | 0.2439 | 0.4154 | 0.0626 |
| Max error | 4.0455 | 1.2520 | 5.9568 | 2.0736 |

Table 14: Performance of the Periodic Set Transformer on prediction of band gap energy across each fold for various error measures.

| Measure | Mean | Min | Max | Std. |
|---|---|---|---|---|
| MAE | 0.2668 | 0.2610 | 0.2742 | 0.0045 |
| RMSE | 0.6043 | 0.5853 | 0.6182 | 0.0117 |
| MAPE | 5.6774 | 3.1799 | 10.368 | 2.6024 |
| Max error | 7.3874 | 7.0041 | 7.8945 | 0.3415 |

be applied to the test set. This is *not* the case in our testing procedure. Instead, we train the model until the validation error has converged and then apply the model after the final epoch to the test set. Datasets with fewer samples, particularly those for phonon peak and refractive index, produce erratic validation errors when the model is trained. For this reason, we keep training the model until this value converges or fluctuations become adequately small. The exact hyper-parameters for each property are included in Table 18.

Table 15: Performance of the Periodic Set Transformer on prediction of last phonon density of state peak across each fold for various error measures.

| Measure | Mean | Min | Max | Std. |
|---|---|---|---|---|
| MAE | 36.698 | 34.736 | 38.957 | 1.8325 |
| RMSE | 68.171 | 55.827 | 81.905 | 10.762 |
| MAPE | 0.0712 | 0.0635 | 0.0790 | 0.0056 |
| Max error | 521.81 | 345.96 | 724.92 | 156.24 |

Table 16: Performance of the Periodic Set Transformer on prediction of exfoliation energy across each fold for various error measures.

| Measure | Mean | Min | Max | Std. |
|---|---|---|---|---|
| MAE | 36.614 | 22.220 | 51.398 | 10.112 |
| RMSE | 98.321 | 37.676 | 149.67 | 45.673 |
| MAPE | 3.8386 | 0.2584 | 17.458 | 6.8113 |
| Max error | 666.81 | 154.85 | 1526.4 | 495.84 |

Table 17: Compute time in seconds for each of the datasets averaged across the five folds.

| Property | Samples | Batch Size | Time per Epoch (s) | Time per Fold (m) |
|---|---|---|---|---|
| Formation Energy | 132,752 | 256 | 84.8 | 353.4 |
| Band Gap Energy | 106,113 | 256 | 63.4 | 264.1 |
| Shear Modulus | 10,987 | 128 | 6.84 | 28.5 |
| Bulk Modulus | 10,987 | 128 | 7.12 | 29.67 |
| Refractive Index | 4,764 | 32 | 3.22 | 10.74 |
| Phonon Peak | 1,265 | 128 | 1.07 | 4.46 |
| Exfoliation Energy | 636 | 32 | 0.53 | 1.76 |

Table 18: Hyper-parameters of the Periodic Set Transformer used for the results in Table 2. Properties in the table are abbreviated like so: phonon peak (PP), refractive index (RI), bulk modulus (BK), shear modulus (SM), band gap energy (BG), formation energy (FE), and exfoliation energy (EE).

| Hyper-parameter | PP | RI | BM | SM | BG | FE | EE |
|---|---|---|---|---|---|---|---|
| Attention Heads | 4 | 4 | 4 | 4 | 8 | 2 | 4 |
| Head Size | 256 | 64 | 256 | 256 | 64 | 256 | 64 |
| Encoders | 4 | 4 | 4 | 4 | 8 | 8 | 4 |
| Dropout | 0.1 | 0.1 | 0.1 | 0.1 | 0 | 0 | 0.1 |
| Attn. Dropout | 0 | 0 | 0 | 0 | 0.05 | 0 | 0 |
| Batch Size | 128 | 32 | 128 | 128 | 256 | 256 | 32 |
| Weight Decay | 3e-5 | 1e-5 | 3e-5 | 3e-5 | 0 | 0 | 3e-5 |
| Epochs | 250 | 200 | 250 | 250 | 250 | 250 | 200 |

