# OpenReview forum: "Periodic Set Transformer: Material Property Prediction from Continuous Isometry Invariants"
_ICLR.cc/2024/Conference — Submitted to ICLR 2024_

### Official Review · Reviewer_yW4J · 2023-10-22

**Soundness:** 3 good
**Presentation:** 2 fair
**Contribution:** 2 fair
**Rating:** 5
**Confidence:** 4

**Summary:**

This paper proposes Pointwise Distance Distribution (PDD), a continuous isometry invariant for periodic point sets for the representation learning of crystals. It develops a transformer model, Periodic Set Transformer (PST), with a modified attention mechanism that integrates composition information and structural encoding for accurate crystal property prediction. By defining the crystal in terms of a periodic set, the representation of crystals encodes the periodicity of crystals and becomes continuous under perturbations, bridging the gap between crystal descriptors and machine learning models. As a result, the transformer model PST equipped with modified self-attention and PDD-weighted readout has the potential to make accurate predictions for lattice energies. Furthermore, the authors extend PST for crystal property predictions, outperforming graph-based or transformer-based models on some tasks, given the extensive experimental results on Matbench. The evidence from ablation studies further proves the effectiveness of the combination of compositional and structural embeddings for a better understanding of the chemical space of crystals.

**Strengths:**

Originality: The paper proposes PDD for the representation of periodic lattice and overcomes the discontinuity of traditional graph representations. Therefore, the paper uniquely contributes to the field by exploring the reasonable representations for periodic sets which can help machine learning models fully learn the geometry of the space.

Quality: The paper carefully explains the core concepts like PDD with detailed derivation. Besides, the extensive experimental results and visualizations provide convincing evidence to support the statement in the paper.

Clarity: The paper effectively communicates its ideas and findings with clarity. The paper is well-written, and the logic is coherent.

Significance: The paper focuses on improving the embeddings for crystals so that the transformer model equipped with the adapted self-attention mechanism could be leveraged for crystal property predictions. The experimental results in the manuscript show the potential of PST model to outperform the widely used graph-based models for crystal property predictions. The model could be further improved by pertaining, making it a promising candidate in crystal property prediction and crystal structure optimization.

**Weaknesses:**

1. Although the authors' presentation is quite clear in general, the details of the experiments provided in the paper are not enough, especially why the experiments are designed in this way, what are the datasets and targets, and what the difference across experiments is and how they collaborate to support the statements in the manuscript.

2. The description in the section Prediction of Lattice Energy is quite vague. For example, the authors do not specify what the datasets (e.g. T2, P1, S2) are, how they are obtained, and what kinds of data entries are included in them. Otherwise, it's hard to figure out why the experimental results here are significant. The author might consider revising this section so that the logic is more transparent to readers.

**Questions:**

1. In terms of the explanation of isometry on page 2, I'm wondering why the isometry has the form $f(S)=Q$ and $g(Q)=S$. From my understanding, isometry means $d_S(a,b) = d_Q(f(a),f(b)), a,b \in Q$, and I can't tell that this is equivalent to the explanation in the manuscript.

2. Earth Mover's Distance is mentioned in the Introduction part, but I do not see detailed descriptions about how to use it for crystal representation in the manuscript. Besides, have the authors considered comparing with ElMD [1], which has also introduced Earth Mover’s Distance as metrics for chemical similarity and inorganic compound embeddings?

3. In the second experiment of prediction of lattice energy, if I'm understanding it correctly, the datasets consist of crystals with different compositions while the compositional information is not included. Then how does the model make predictions for two similar lattices with different compositions? And even if the model can outperform the baseline on this task, I'm afraid it cannot demonstrate that the model is applicable to practical usage.

4. Why is the PST model evaluated on the training set for the first two tasks of lattice energy prediction? And what is the reason for supplementing P2M data to training data to reduce error? From the results in Table 1 & 2, I cannot be persuaded of the effectiveness of PST.

5. Could you clarify how the contribution is calculated in the ablation study? And I think the errors here are sufficient to demonstrate the impact of compositions and PDD.

[1] Hargreaves, C. J., et. al., The earth mover’s distance as a metric for the space of inorganic compositions. Chemistry of Materials, 32(24), 10610-10620, 2020.

---

> ### Author Response · Authors · 2023-11-12
> **Response to Reviewer yW4J**
>
> Thank you for the feedback, it has allowed us to address several issues in the manuscript.
>
> > the details of the experiments provided in the paper are not enough..
>
> The intention of the two experiments is to illustrate the flexibility of the model and the effectiveness of using invariants outside of their usual context of crystal comparison and instead in representations for ML.
>
> In the first case, the datasets used in section 4 and section 5 are significantly different. Section 4 uses molecular crystals with each set (e.g. T2) containing the same composition but changes in structure while the MP crystals are mostly inorganic crystals with varied compositions.
>
> In the second case, the PDD and PST are able to perform comparatively to graph-based models that contain additional structural information. The graph-based models also use $k$-NN distances but through the use of edges, they can specify between which atoms a distance corresponds to. The PDD does not have this information, it only uses the distances themselves. Despite this, the PST still outperforms CGCNN (the most comparable model in terms of information used) in 5/6 properties. The use of edges is discontinuous under atomic perturbations. If a small change in atomic positioning occurs, the set of $k$-NN can change. In a graph-based model, this means edges can shift, changing the graph's topology in a way that is disproportionate to a small change in atomic position. In the case of the PDD, when this occurs the distances will be changed continuously because there is no reference to the neighboring atoms.
>
> > The description in the section Prediction of Lattice Energy is quite vague..
>
> The primary purpose of the PDD is to distinguish two crystals via the EMD. Its efficacy in this can be seen in the MDS plots (Figure 4a) contained in the appendices. The question this experiment seeks to answer is whether or not this means it is also an effective representation for crystals in an ML context. We will add further context to the revision.
>
> >  I'm wondering why the isometry has the form..
>
> In the context of the explanation $f: Q \rightarrow S$ and $g : S \rightarrow Q$ are bijective isometries and should the condition $f(Q) = S$ and $g(S) = Q$ be met, then the two periodic point sets $S$ and $Q$ are isometric. This latter part could be generalized to $S$ and $Q$ being metric spaces in $\mathbb{R}^n$. In our explanation, we don't directly define an isometry and instead mention it is a distance-preserving mapping between metric spaces; this will be adjusted in the revision.
>
> > Earth Mover's Distance is mentioned in the Introduction part..
>
> The Earth Mover's Distance establishes a continuous metric on periodic sets and is used to measure differences in PDDs. It is not used to create features or utilized in the model. It is used in plots 4a and 4b in the appendix to show how distances between structures can be a useful indicator for differences in their property values.
>
> > how does the model make predictions for two similar lattices with different compositions?
>
> It is true the compositional information is not included (in section 4). The experiment shows that the structural differences produced from having different compositions are reflected in the distances in the PDD and this is enough to make accurate predictions. This is already shown to be the case when distances are measured via the EMD and then projected onto 2D space (Figure 4a of the appendix). Whether this is true when the PDD is input to an ML algorithm is what this experiment aims to address.
>
> > Why is the PST model evaluated on the training set for the first two tasks of lattice energy prediction?
>
> The table was trying to indicate what crystals make up the training and test sets. The reported error is for the test set. The first task follows the same setup as Ropers et al. The second extends this to multiple crystal sets.
>
> > And what is the reason for supplementing P2M data to training data to reduce error?
>
> We tried to show that the model can be improved with a small amount of generated data for a new crystal type (P2M) which has not been seen by the model before. In a practical setting, DFT calculations (computationally expensive) can be done on a small number of crystals and be added to the training set. These additional crystals can improve the prediction accuracy on this new type of crystal. The PST can then be applied during the rest of the CSP process in which (tens of) thousands of additional structures could be generated.
>
> > Could you clarify how the contribution is calculated in the ablation study?
>
> If $e_{PDD}$, $e_{Comp}$ and $e_{PST}$ are the mean-absolute-errors for the contribution for the PDD component would be $c_{PDD} = r_{PDD} / (r_{PDD} + r_{Comp})$ and the composition contribution would be $c_{Comp} = r_{Comp} / (r_{PDD} + r_{Comp})$
>     where $r_{PDD} =  e_{PST} /  e_{PDD}$ and $r_{Comp} = e_{PST} / e_{Comp}$.
>
> Please let us know if further explanation is needed.

---

> > ### Comment · Reviewer_yW4J · 2023-11-14
> >
> > Thanks for the author's response to my questions and concerns. I really appreciate that you have helped me better understand your contributions.
> >
> > The responses have almost resolved my concerns; however, I do have further questions:
> >
> > -- The description of the datasets are still vague: the datasets T2, P1, S2, P1M, P2, and P2M seem to come out suddenly and the readers only have a very general idea about what they are about. If those datasets come from literature, they should be cited in the manuscript; if the authors curated the datasets, how the datasets were curated should be mentioned. Besides, basic information like data size should also be summarized.
> >
> > -- The authors mentioned in the rebuttal that "In our explanation, we don't directly define an isometry and instead mention it is a distance-preserving mapping between metric spaces; this will be adjusted in the revision". However, if that is the case, I think the model here is an invariant neural network and I don't understand how is isometry actually encoded.
> >
> > -- Further questions about results on Matbench: could you also include the error bars in Table 3? And how's the data efficiency and gpu memory cost compared with baseline models (or even other models on Matbench Leaderboard like coNGN, ALIGNN, or MODNET)?
> >
> > Hope to see your comments on my further questions! Thanks.

---

> > > ### Author Response · Authors · 2023-11-16
> > >
> > > Thank you for getting back to us, it is appreciated.
> > >
> > > > The description of the datasets are still vague: the datasets T2, P1, S2, P1M, P2, and P2M seem to come out suddenly..
> > >
> > > We have included more context in this section so the transition is not so rough.
> > >
> > > >  If those datasets come from literature, they should be cited in the manuscript
> > >
> > > We cite the work that they were generated from [1] like so:
> > > > Prediction of lattice energy is done in three different scenarios using experimentally generated crystals from Pulido et al. (2017)
> > >
> > > but any further details are admittedly sparse. We will include more details on generation/acquisition and the characteristics of the dataset.
> > >
> > > [1] Functional materials discovery using energy-structure-function maps. Nature, 543(7647):
> > > 657–664, March 2017
> > >
> > > > Besides, basic information like data size should also be summarized.
> > >
> > > Agreed, these will be included in Table 1 in the revision.
> > >
> > > > However, if that is the case, I think the model here is an invariant neural network and I don't understand how is isometry actually encoded.
> > >
> > > So the PDD is an isometry invariant. That is, if an isometry falling under rigid motion is applied to the atomic coordinates then the PDD of the corresponding periodic point set remains unchanged. The model is not necessarily invariant to these isometries, 3D coordinates, for example, could be used as input if desired and these are not invariant under rotation, translation etc.
> > >
> > >
> > > > could you also include the error bars in Table 3?
> > >
> > > Could you specify what exactly you mean by this?
> > >
> > > > And how's the data efficiency and GPU memory cost compared with baseline models (or even other models on Matbench Leaderboard like coNGN, ALIGNN, or MODNET)?
> > >
> > > coGN/coNGN and MODNET have package dependencies that require a CUDA version higher than our GPU allows and the matbench code for ALIGNN does not actually do any training (it takes results from a csv). We think the code could be refactored to allow use for our GPU, but this would likely take some time so to give you some sense of computational cost we used a model that was ready to run. Additionally, we do not currently have access to a GPU with large enough memory to run Band gap and formation energies. The results for PST are re-run for the current hardware so things are comparable. The time per epoch is determined by taking the average of a 100 epoch run without pre-processing time or validation time. Memory was measured using nvidia-smi. All results are run on a Nvidia GTX 1060 with 6GB of memory. The "epochs" column indicates the original number of epochs the model was meant to run to achieve its reported results.
> > >
> > > By property for MEGNet:
> > >
> > >
> > > Bulk Modulus:
> > >
> > > |Model| GPU Memory Usage (MB) | Time per Epoch (s) | Epochs | Batch Size |
> > > |-----|-----------------------|--------------------|--------|------------|
> > > |MEGNet| 847 | 8.16 | 1000 | 32 |
> > > |PST| 3875 | 7.71 | 250 | 128 |
> > >
> > > Shear Modulus:
> > >
> > > |Model| GPU Memory Usage (MB) | Time per Epoch (s) | Epochs | Batch Size |
> > > |-----|-----------------------|--------------------|--------|------------|
> > > |MEGNet| 847 | 8.21 | 1000 | 32 |
> > > |PST| 3875 | 7.68 | 250 | 128 |
> > >
> > > Refractive Index:
> > >
> > > |Model| GPU Memory Usage (MB) | Time per Epoch (s) | Epochs | Batch Size |
> > > |-----|-----------------------|--------------------|--------|------------|
> > > |MEGNet| 1338 | 4.96 | 1000 | 32 |
> > > |PST| 951 | 3.60 | 200 | 32 |
> > >
> > >
> > > Phonon Peak:
> > >
> > > |Model| GPU Memory Usage (MB) | Time per Epoch (s) | Epochs | Batch Size |
> > > |-----|-----------------------|--------------------|--------|------------|
> > > |MEGNet| 583 | 0.874 | 1000 | 32 |
> > > |PST| 1069 | 0.675 | 250 | 128 |
> > >
> > > Exfoliation Energy:
> > >
> > > |Model| GPU Memory Usage (MB) | Time per Epoch (s) | Epochs | Batch Size |
> > > |-----|-----------------------|--------------------|--------|------------|
> > > |MEGNet| 463 | 0.655 | 1000 | 32 |
> > > |PST| 603 | 0.52 | 200 | 32 |
> > >
> > > Please let us know if you have further questions.

---

> > > > ### Comment · Reviewer_yW4J · 2023-11-19
> > > >
> > > > Thanks for your answers.
> > > > -- I mean, in Table 2 (sorry I referred to the wrong table), since you provide the MAE results on Matbench with 5-fold cross-validation, why not also include the standard deviation of the results from each fold which you can find in Matbench for other models?
> > > > -- It seems that PST costs more memory but could be trained faster compared with MEGNet. Interesting results. I came up with this question because there are more important things than just providing results on benchmark: either significantly improving the leaderboard (not just barely becoming the SOTA) or having some practical applications. The latter could be accelerating the discovery of new materials (not just talking about it), achieving comparable performance while becoming more efficient, etc. This is slightly beyond the scope of review for a conference but it'll be great if the authors can really look into it.

---

> > > > > ### Author Response · Authors · 2023-11-20
> > > > >
> > > > > > why not also include the standard deviation of the results from each fold which you can find in Matbench for other models?
> > > > >
> > > > > Thank you for clarifying. It was just a matter of practicality considering limitations on space. We opted to include more models in the comparison. It would be useful to have those results as well so we will add them to the supplementary material.
> > > > >
> > > > > > It seems that PST costs more memory but could be trained faster compared with MEGNet.
> > > > >
> > > > > It is worth noting memory consumption is tied pretty heavily to batch size. We used the batch size that provided the results in Table 2 for consistency.
> > > > >
> > > > > > I came up with this question because there are more important things than just providing results on benchmark: either significantly improving the leaderboard (not just barely becoming the SOTA) or having some practical applications. The latter could be accelerating the discovery of new materials (not just talking about it), achieving comparable performance while becoming more efficient, etc.
> > > > >
> > > > > We appreciate the feedback. It will be something we look into going forward, thank you.

---

> > > > > > ### Comment · Reviewer_yW4J · 2023-11-22
> > > > > >
> > > > > > Thanks a lot for the further explanations and great insight!
> > > > > >
> > > > > > Given the response to my concerns and criticisms by other reviewers, I'll not raise my score. Wish you all the best in the future.

---

### Official Review · Reviewer_DofE · 2023-10-28

**Soundness:** 2 fair
**Presentation:** 2 fair
**Contribution:** 2 fair
**Rating:** 3
**Confidence:** 4

**Summary:**

The paper proposes a new representation for machine learning on crystal structures based on Point Distance Distribution (PDD), which the paper claims is both continuous and isometric. The proposed representation augments augments PDD with composition in order to be able to represent a crystal in a unique manner such that machine learning models can be applied to it. The paper also proposes a modified self-attention mechanism that can utilize the PDD and compositional information to predict a variety of materials properties.

The paper starts by introducing crystal structures and their associated challenges of predicting their properties that traditional computational chemistry methods that are often computationally prohibitive in evaluating properties for many materials. Next, the paper describes the challenge of finding good representations for crystal structures that are isometric and machine learning friendly and defines a set of properties that a good representation should have including invariance, completeness, and continuity. Following a description of related work, the paper discusses the PDD and their proposed periodic set transformer including detailed mathematical definitions. Next the paper describes the PDD encoding that incorporates atom composition information and how it is incorporated in the periodic set transformer.

Following the definition of the method, the paper provides two case studies: one for lattice energy prediction and one for materials property prediction based on Materials Project. In the lattice energy prediction study, the paper investigates the effects of different methods with PDD generally showing better performance. In the case of materials property predictions for Materials Project, the results are more mixed with other methods outperforming PST in some, but not all, cases. The paper then provides an ablation study mostly focusing on the input  representation for Materials Project property prediction followed by the conclusion summarizing the work.

**Strengths:**

The paper has the following strengths:
* The paper provides a new representation for machine learning on crystal structures that has very useful properties, including isometry and continuity. The representation itself could be promising for the development of other machine learning methods (originality, significance).
* The paper provides a new attention mechanism tailored to the PDD representation, which is then applied to different case studies with some results indicating the utility of the representation and the architecture (originality).

**Weaknesses:**

While the paper introduces an interesting and relevant idea, the current form includes some major weaknesses:
* The description of the PDD representation and the architecture is often unclear and confusing (clarity).
* The contribution appears limited to the inclusion of the composition on top of the PDD representation, which appears to be prior work (significance, originality).
* The experiments performed are relatively small in scale with the results often not well presented (clarity, quality).
* The experiments are in Section 4 are not well described making it difficult to assess their significance (clarity, quality). Given that only PDD representations were used, it is unclear what contributions of the paper are being highlighted here. Also the model architectures used are unclear. My best guess is that it involves Gaussian processes similar to the AMD case.
* Many of the figures and tables are only sparsely labeled making it difficult to fully understand the takeaways. (clarity)
* The notation in Section is hard to follow given that there are letters in upper and lower case with different bold fonts each corresponding to different entities. This can be improved for greater clarity.

**Questions:**

* What are the model architectures used Section 4?
* Can you describe in more details how the rows of the PDD representation are collapsed into each, specifically how identical rows are identified?
* How are the rows of the PDD representation ordered? Does this ordering matter?
* How do atoms get counted in the PDD construction described in Section 3.1? Since composition is not present yet, are the atoms indexed without atom types?
* Is there a predetermined way to choose k for the PDD? Based on the information in the appendix it appears to be a hyperparameter that seems significant. It would be good to more details on this.
* What types of crystals are studied in Section 4? You mention both molecules and crystals here, so are these molecular crystals? Is there a reason you claim that only the lattice matters for these structures? Clarity could be substantially improved by providing more detailed information about the task.
* In Section 5 - is there a reason that CrabNet cannot use PDD embeddings? I would assume that GNNs use a different representation in your study, which would also be good to clarify.

---

> ### Author Response · Authors · 2023-11-12
> **Response to Reviewer DofE**
>
> Thank you for your feedback. It seems that clarity was an issue, we will try to make things more clear in the revision. In response to your comments/questions:
>
> > The description of the PDD representation and the architecture is often unclear and confusing (clarity).
>
> We have tried to supplement the definition of the PDD with an example in the appendix. If you could specify what about the explanation of architecture and PDD representation was unclear that would be greatly appreciated.
>
> > The contribution appears limited to the inclusion of the composition on top of the PDD representation..
>
> The PDDs original purpose was to measure differences in the structure of crystals in order to compare them. We believe we have provided evidence that it (and potentially other invariants) has a place in machine learning as the input representation. Additionally, the model we propose can be applied to data with the structure of a weighted set (or discrete distribution). The method of turning multisets into weighted sets by grouping inputs based on their features and using weights to describe multiplicity can be applied to similar Transformer input to decrease the number of tokens in the input, improving computational performance. For the experiments in Section 5 the size of the set of the input is reduced to $\sim 45$ percent of the original number of atoms in the unit cell.
>
> > The experiments performed are relatively small in scale ..
>
> Could you please expand on this? Several other models report between 1-3 datasets including CGCNN  (only the Materials Project), MEGNet (the Materials Project and QM9, which is a molecular not crystal dataset), and SchNet (3 molecular datasets). We will be adding an additional ablation study and the results on the effect of the collapse tolerance on input size compared to other graph approaches (based on another reviewer's comments).
>
> > Given that only PDD representations were used, it is unclear what contributions of the paper are being highlighted here..
>
> A similar critique has been mentioned by other reviewers, we will be more explicit about the setup and purpose of the experiment in section 4 of the revised version of the manuscript. The comparison done in this experiment is between the PST using only the PDD and Gaussian regression with AMD as input.
>
> > What are the model architectures used Section 4?
>
> This model uses the Transformer described in section 3.1, but instead of using PDD encoding to incorporate structural information with compositional information, it uses just the PDD alone. The model we compare to uses the AMD as the representation for a crystal and the algorithm applied is Gaussian regression.
>
> > Can you describe in more detail how the rows of the PDD representation are collapsed into each, specifically how identical rows are identified?
>
> Two rows are identified as identical if their distance is less than or equal to the collapse tolerance using a valid distance metric (definition 3.2). In our application, we use the $L_\infty$ metric. These two rows are collapsed into one by taking the average of the rows.
>
> > How are the rows of the PDD representation ordered? Does this ordering matter?
>
> They are ordered lexicographically (definition 3.2). The order does not matter as the attention mechanism is permutation equivariant and the pooling layer is permutation invariant.
>
> > How do atoms get counted in the PDD construction described in Section 3.1?..
>
> The number of atoms in the unit cell defines their count. Through collapsing, however, which unit cell is used does not matter. The PDD contains no knowledge of the composition, only the pairwise distances.
>
> > Is there a predetermined way to choose k for the PDD?
>
> $k$ is selected to be large enough such that the generic completeness property of the PDD is satisfied. That is, the distances in the final column of the PDD are greater than twice the covering radius of the lattice. This varies from crystal to crystal so $k$ is selected such that it is adequate for 99\% of the crystals in the largest dataset.
>
> > What types of crystals are studied in Section 4?...
>
> Yes, they are molecular crystals. Specifically, simulated crystals created during the crystal structure prediction process. For each crystal set (e.g. T2) the crystals share the same composition but have different structure.
>
> The PDD, is used for measuring differences in structure via EMD. Figure 4a in the appendix shows how the pairwise distances manifest on a 2D plane, creating clusters by composition. The purpose of the experiment is to determine whether such an inference can be made in a ML application wherein the PDD is the input.
>
> > is there a reason that CrabNet cannot use PDD embeddings?
>
> Not necessarily. The fractional encoding they use is determined by the proportion of each element in the material. This would need to be changed to account for two atoms of the same element that have differing $k$-NN distances.
>
> Please let us know if anything is unclear.

---

> > ### Comment · Reviewer_DofE · 2023-11-18
> >
> > Thank you for comments and updates. After going through your responses and updated draft, I have some additional questions and comments:
> >
> > -  **Experiment Scale:** The current scope of Materials Project, which is the basis for MatBench, has ~160k samples of crystal structures. There are other datasets in the open literature with greater diversity of crystals (e.g OQMD [1], NOMAD [2]) that would help make a stronger case for the capabilities of your proposed method. The Open MatSci ML Toolkit [3] provides further details and a way to interact with these additional datasets.
> > - **Transformer Model Comparisons:** In your ablation in Section 5.2, it would be good to see additional results of transformer based methods (e.g. CrabNet) directly. Ideally, one would also the new representation in those architectures as well, but that might come with additional challenges. In that case, it would be good to include such details in the paper itself.
> > - **Ablation Study:** Thank you for including the ablation study in Section 5.2; this provides important additional details.
> > - **Limitations:** I would like to see a more extensive discussion of limitations of the proposed PDD encoding. One question to ask, for example, would be whether the encoding can only manage periodic structures and/or if it can be adjusted to include surfaces and more complex materials systems. Understanding this can be helpful for diverse applications, such as catalysts (e.g. OpenCatalyst Project [2]) and more complex materials systems.
> >
> > [1] Kirklin, S., Saal, J. E., Meredig, B., Thompson, A., Doak, J. W., Aykol, M., ... & Wolverton, C. (2015). The Open Quantum Materials Database (OQMD): assessing the accuracy of DFT formation energies. npj Computational Materials, 1(1), 1-15.
> >
> > [2] Draxl, C., & Scheffler, M. (2019). The NOMAD laboratory: from data sharing to artificial intelligence. Journal of Physics: Materials, 2(3), 036001.
> >
> > [3] Lee, K. L. K., Gonzales, C., Nassar, M., Spellings, M., Galkin, M., & Miret, S. (2023). MatSciML: A Broad, Multi-Task Benchmark for Solid-State Materials Modeling. arXiv preprint arXiv:2309.05934.
> >
> >
> > [4] Chanussot, L., Das, A., Goyal, S., Lavril, T., Shuaibi, M., Riviere, M., ... & Ulissi, Z. (2021). Open catalyst 2020 (OC20) dataset and community challenges. Acs Catalysis, 11(10), 6059-6072.

---

> > > ### Author Response · Authors · 2023-11-20
> > >
> > > Thank you for getting back to us, it is appreciated.
> > >
> > > >  The current scope of Materials Project, which is the basis for MatBench, has ~160k samples of crystal structures. There are other datasets in the open literature with greater diversity of crystals (e.g OQMD [1], NOMAD [2]) that would help make a stronger case for the capabilities of your proposed method. The Open MatSci ML Toolkit [3] provides further details and a way to interact with these additional datasets.
> > >
> > > Thank you for suggesting additional datsets. We will try to add them to the results.
> > >
> > > > In your ablation in Section 5.2, it would be good to see additional results of transformer based methods (e.g. CrabNet) directly. Ideally, one would also the new representation in those architectures as well, but that might come with additional challenges. In that case, it would be good to include such details in the paper itself.
> > >
> > > While our approach and the approach of CrabNet aren't necessarily contradictory, the combining  of their fractional encoding with PDD encoding would differentiate atoms by both their $k$-NN and their frequency in the crystal. After this is done, the results which justify the use of PDD encoding (invariance and generic completeness of periodic points sets) cannot be guaranteed. The method for combining both encodings would have to be something different than what both we and CrabNet propose.  We will add this to the discussion of the results section.
> > >
> > > > One question to ask, for example, would be whether the encoding can only manage periodic structures and/or if it can be adjusted to include surfaces and more complex materials systems.
> > >
> > > The encoding method maintains its properties when applied to periodic sets in $R^n$ so invariance and generic completeness can be guaranteed for periodic crystals and quasi-periodic crystals that can be mapped as periodic sets in higher dimensions. Additionally, it retains these properties on finite points clouds in $R^n$, so application of the encoding could be extended to molecules, proteins, etc. Beyond this however, the theoretical results cannot be guaranteed. Though, this does not necessarily prohibit the application of our method. We will be sure to include these limitations in the article.
> > >
> > > Thank you for the feedback and please let us know if you have any more questions or comments.

---

### Official Review · Reviewer_V7SM · 2023-10-31

**Soundness:** 2 fair
**Presentation:** 2 fair
**Contribution:** 2 fair
**Rating:** 3
**Confidence:** 5

**Summary:**

The authors introduce a transformer model with a modified self-attention mechanism that adapts PDD (Pointwise Distance Distribution, represented by a k-nearest-neighbor distance matrix), and incorporates compositional information via a spatial encoding method. Specifically, the authors consider the PDD as a set of grouped atoms and use an attention mechanism to find interactions between members of the set. The authors claim that PDD effectively distinguishes periodic point sets up to isometry but doesn't consider the composition of the underlying material, and thus, the newly proposed encoding method can effectively capture this information.

**Strengths:**

The introduction of the Periodic Set Transformer (PST) model is articulated in a straightforward manner. The authors have designed the PST model to incorporate not just structural but also compositional information through Pointwise Distance Distribution (PDD) Encoding. This makes the model versatile and potentially more effective in predicting material properties.

**Weaknesses:**

Majors:

1. **Inadequate Experimental Results**: The paper's experimental section reveals suboptimal performance in predicting key electronic properties of crystals, such as formation energy. Notably, the proposed method performs poorly in comparison to other methods listed in the results table. Furthermore, the paper lacks a comparison with the state-of-the-art method coGN [1] at Matbench, which is a significant oversight.
2. **Lack of Novelty in k-Nearest-Neighbor Construction**: The paper does not sufficiently differentiate its approach from k-nearest-neighbor graph construction of message-passing methods. Common methods for material prediction, such as  CGCNN [3] and ALIGNN [4], also consider both atomic properties and distances, raising questions about what exactly is the authors’ method beyond those message-passing methods with k-nearest-neighbor graph construction.

Minor:
1. **Omission of Citations**: The authors don't include important baselines coGN [1] and PotNet [2].


Ref:

[1] Ruff, Robin, et al. "Connectivity Optimized Nested Graph Networks for Crystal Structures." *arXiv preprint arXiv:2302.14102* (2023).

[2] Lin, Yuchao, et al. "Efficient Approximations of Complete Interatomic Potentials for Crystal Property Prediction." *ICML 2023*.

[3] Xie, Tian, and Jeffrey C. Grossman. "Crystal graph convolutional neural networks for an accurate and interpretable prediction of material properties." *Physical review letters* 120.14 (2018): 145301.

[4] Choudhary, Kamal, and Brian DeCost. "Atomistic line graph neural network for improved materials property predictions." *npj Computational Materials* 7.1 (2021): 185.

**Questions:**

See weeknesses.

---

> ### Author Response · Authors · 2023-11-12
> **Response to Reviewer V7SM**
>
> Thank you very much for the review, the feedback will allow us to improve the quality of the article.
>
> In response to the concerns you raised:
>
> > The paper's experimental section reveals suboptimal performance in predicting key electronic properties of crystals, such as formation energy
>
> While this is true, it still performs well on other properties in comparison to other similar graph-based models. Additionally, our model outperforms the other Transformer model we compare to (and the only one on Matbench) in 5/6 properties.
>
> > Furthermore, the paper lacks a comparison with the state-of-the-art method coGN
>
> While it is true that line graph-based methods such as ALIGNN and coGN perform with higher accuracy, they also incur a significantly higher computational cost. In the crystal graph construction proposed by CGCNN and used in ALIGNN, if a crystal contains $n$ atoms in the unit cell, the resulting graph will also have $n$ vertices. Each vertex will have $k$ directed edges assuming a large enough cutoff radius (which is suggested to prevent isolated portions of the graph). The line graph for this graph would then contain $k$ vertices and $\frac{n}{2} (k^2 - k)$ edges. coGN adds an additional line graph to account for dihedrals. For a crystal with just 4 atoms in the unit cell, the line graph used in ALIGNN for the angles with $k=12$ would have 48 nodes and 264 edges. The dihedral line graph would contain 264 vertices and 17,424 edges. The crystals in the materials project dataset on average contain $\sim 24$ atoms in the unit cell. In contrast, the approach we propose decreases the size of the representation by grouping similar atoms together. We have mentioned how the collapse tolerance affects the cardinality of the input set in response to another reviewer, we include it here as well. The size of our representation compared to the number of atoms in the unit cell at a collapse tolerance of 1e-4 is as follows:
>
> Phonon Peak: 0.464
>
> Formation energy: 0.444
>
> Band Gap: 0.437
>
> Bulk Modulus: 0.414
>
> Shear Modulus: 0.414
>
> Refractive Index: 0.434
>
>  where on average if a dataset has 24 atoms in the unit cell, our representation's average size would be roughly  $24 \times 0.464 \approx 11$ (in the case of phonon peak).
>
> > The paper does not sufficiently differentiate its approach from k-nearest-neighbor graph construction of message-passing methods
>
> Graph construction in other work often needs the use of a cutoff radius to prevent ambiguous graph construction between neighbors with the same distance. This is not needed in our case as we only use the distance and have no reference to the neighbor's atomic type. This radius and the value of $k$ are often not justified in their selection and are instead based on trial and error. We propose a heuristic for finding an adequate value of $k$ based on the theoretical results of generic completeness for the PDD. This is mentioned in the experiments in the appendix, but it seems pertinent so it will be moved to the main body.
>
> Graph-based methods also use edges to indicate between which atoms the distances lie. This provides additional structure to the data but at the cost of being discontinuous under atomic perturbations. Small changes in the position of atoms can change the set of atoms in the set of $k$-NN. If this occurs edges will shift, changing the graph's structure. In our representation we use the same information but in a different way. Despite not using any references to neighboring atoms to indicate which distances correspond to which neighboring atoms, our model still performs on par or better than similarly informed graph-based models on several properties. Additionally, because only distances to the neighbors are used, perturbations will change the distances in a continuous manner.
>
> > The authors don't include important baselines coGN [1] and PotNet [2]
>
> These two will be added to the review of related work and the previous rationale in the first point will be included in the section which discusses the selection of models to compare our model to.
>
> Please let us know if you have further concerns or if anything we have mentioned is unclear.

---

> > ### Comment · Area_Chair_SXmV · 2023-12-02
> > **Does the response address your concerns?**
> >
> > @Reviewer V7SM,
> >
> > I would appreciate it if you could review the response and adjust your feedback (and rating) as necessary.
> >
> > AC

---

> ### Comment · Reviewer_V7SM · 2023-12-03
>
> Hi AC, reviewers, and authors,
>
> I keep my current rating for the reasons below -
>
> First, for the used dataset Material Project, formation energy is the most important property for evaluation, as it involves more samples compared with the other properties. If there is a claim that - the method is specifically designed to deal with a particular property, then it makes some sense. However, this is not the case in this paper.
>
> For the method, especially compared with KNN-based GNNs, the motivation is good. However, the statements in the paper are obscure; the paper was not updated to incorporate the comments from reviewers; and again, the evaluation is not convincing to support the claim.
>
> I suggest authors pay more attention to the formation energy in empirical studies, and revise the paper significantly to incorporate the comments of reviewers.

---

### Official Review · Reviewer_kc8a · 2023-11-06

**Soundness:** 3 good
**Presentation:** 3 good
**Contribution:** 2 fair
**Rating:** 6
**Confidence:** 3

**Summary:**

The Pointwise Distance Distribution (PDD) is a recently developed invariant for periodic crystals that is easy to compute and differentiates between almost all non-isomorphic lattices (generically complete). The authors of the present work propose to use the PDD in a transformer architecture to learn to predict the lattice energy and other material properties.

The PDD computes for each atom in the unit cell the $k$-nearest neighbour distances and sorts these into a list. Stacking the list for all $n$ atoms gives a $n \times k$ matrix. The PDD is the distribution over these rows, so a discrete distribution over $[0,\infty)^k$. This is an invariant, generically complete, and Lipschitz wrt the earthmover distance on the distributions. The PDD can also be represented as a matrix with weighted rows, where similar rows are collapsed, adding up the weight.

The authors propose to incorporate the PDD data in four ways into a transformer:
- instead of using the atoms as tokens, it uses the rows of the PDD matrix, so collapsing atoms with similar $k$-NN distances
- the initial features are the $k$-NN distances, combined with atomic properties (the authors don't collapse different atoms with similar $k$-NN distances)
- The self-attention is additionally weighted by the PDD weights
- The transformer output is pooled using the PDD weights

The authors show in their experiments that using the PDD is superior to using an alternate invariant, and the authors show that their method performs competitively to other material property prediction methods.

**Strengths:**

- I think it's great to incorporate the powerful PDD invariant into neural networks
- The authors show strong performance on the material prediction dataset.

**Weaknesses:**

- I think an important ablation is missing: just using a typical transformer on the atoms as tokens with the $k$-NN distances and the atomic properties as features. The "PDD" ablation study still only uses the PDD in all the four ways I listed in my summary. It'd be great if the authors could ablate these separately. The CGCNN baseline uses the $k$-NN distances as features, but is not a transformer, so is not a substitute to this ablation.
- A key property of the PDD is its Lipschitz continuity, making it robust to perturbations in the positions. The way the authors use the $k$-NN distances with the hard collapse, then treating the rows as separate tokens, loses this property. Currently, however, the authors are suggesting that the continuity of the PDD is a benefit to their method. The authors should clarify that.

**Questions:**

- In their description of the transformer, it appears like each block only uses self attention and normalization. Is there no MLP used in each block, as is typical in a transformer?
- Could the authors comment on how often the rows of the PDD are collapsed in practice, so how much it matters that the used tokens are aggregates, rather than individual atoms?
- In Def 3.1, the numbers $c_i$ are said to be integer and contained in $[0, 1)$. This would imply they are zero, which I suppose is not what is intended. Could the authors clarify?

---

> ### Author Response · Authors · 2023-11-12
> **Response to Reviewer kc8a**
>
> Thank you for the feedback, it has allowed us to add additional experimental data and clarify definitions in the manuscript. In response to the points you raised:
>
> > I think an important ablation is missing...
>
> In the first scenario you mentioned where collapsing is not done the result should be the same as the PDD-weighted version if the collapse tolerance is exactly zero. The difference in performance compared to higher collapse tolerances is included in the appendices. The representation can be significantly larger than without collapsing and will change should the unit cell chosen be different. In the second scenario, we believe this is equivalent to the "composition" row in the current ablation study where PDD encoding is not used. We are in agreement that in the third and fourth scenarios there should be another ablation to show their effect. Experiments are currently being run and the results will be included in the updated article.
>
> > A key property of the PDD is its Lipschitz continuity..
>
> During the collapse of any two rows, the newly produced row is created taking the average of the two rows. This way even if the collapse tolerance is greater than zero rows are not selected in an either/or manner. This detail was left out, thank you for pointing it out. It will be added to the updated manuscript.
>
> >  Is there no MLP used in each block, as is typical in a transformer?
>
> There is a single layer perceptron which the embeddings are passed through directly after the first layer normalization and then layer normalization is again applied on the sum of the newly updated embeddings (the output of the SLP) and the embedding before it is passed into the SLP.
>
> > Could the authors comment on how often the rows of the PDD are collapsed in practice..
>
> This is dependent on the dataset and the collapse tolerance used. For a collapse tolerance of 1e-4 (the same as used in the main Materials Project results) the number of grouped tokens in the representation over the number of atoms in the unit cell, averaged for each dataset is:
>
> Phonon Peak: 0.464
>
> Formation energy: 0.444
>
> Band Gap: 0.437
>
> Bulk Modulus: 0.414
>
> Shear Modulus: 0.414
>
> Refractive Index: 0.434
>
> We will add information to the article to show how this changes depending on other collapse tolerances.
>
> >  This would imply they are zero, which I suppose is not what is intended. Could the authors clarify?
>
> This portion was indeed unclear, it has been changed such that the lattice keeps its original definition, but the definition of the unit cell has been changed to the space spanned by the parallelepiped $U = { \sum_{i=1}^{n} t_i \mathbf{v}_i | t_i \in [0, 1) }$.
>
> Please let us know if you have further questions.

---

> ### Comment · Reviewer_kc8a · 2023-11-13
> **Question regarding Lipschitz continuity**
>
> Dear authors,
>
> Thanks for your response. I've got a follow-up question regarding the Lipschitz continuity property.
> The PDD is a map between metric spaces from (pointcloud, Euc metric) to (distribution, Earth mover metric) that's Lipschitz continuous. However, in your case, you treat the PDD as a matrix instead of a distribution of rows, so the output shape depends in the input in a non-continuous way. From the construction of the transformer afterwards, it appears to me that the network as whole is not Lipschitz-continuous (or even continuous) when we move a pair of atoms in/out of the $\epsilon$ ball.
> We can see that from the following: the network seems not invariant to duplicating an atom, while halving its weight. It may be that the attention rule in the equation of $r^{(1)}_i$ was designed to have this property, but I don't think that's satisfied due to the weight $w_j$ being outside of the softmax, breaking linearity.
>
> Lipschitz-continuity is a desirable (and as far as I know typically satisfied by most architectures that don't cluster), but not necessary property of such networks. However, I do think that the paper should fairly reflect that this continuity property of PDD is not used in this architecture.
>
> Edit: just to add, as the title contains "continuous" I think this is quite an important discussion.

---

> > ### Author Response · Authors · 2023-11-16
> >
> > > However, in your case, you treat the PDD as a matrix instead of a distribution of rows
> >
> > If you are referring to the computations involved in the Transformer, then this is just for the sake of computational efficiency. The interaction captured by the attention mechanism will be the same for any two tokens (rows in the matrix) regardless of their position in the matrix.
> >
> > > From the construction of the transformer afterwards, it appears to me that the network as whole is not Lipschitz-continuous
> >
> > Yes, this is correct, Dot-Product self-attention is not Lipschitz-continous. We will be sure to change any text that implies otherwise.
> >
> > > ..the network seems not invariant to duplicating an atom, while halving its weight
> >
> > When you say this, are your referring to taking a given PDD and undoing the collapse of two rows? For example, given a PDD of the form:
> >
> > $$\begin{bmatrix}
> > w_1 & a & b\\\\
> > w_2 & c & d
> > \end{bmatrix}$$
> >
> > and expanding it to:
> >
> > $$\begin{bmatrix}
> > w_1 & a & b\\\\
> > w_2 / 2 & c & d\\\\
> > w_2 / 2 & c & d
> > \end{bmatrix}$$
> >
> > If this is the case, the output of the Transformer will be the same for both inputs. If you mean adding an additional atom to the unit cell then no, the output will change. Additionally, there was a mathematical error in the formula in the paper where $V_i$ is included before the PDD weights rescale the attention weights. We have adjusted the notation as well since the previous way it was written mixed individual vector embeddings with the matrix containing all embeddings. Other reviewers found it inconsistent so hopefully this version is more clear. The corrected version for updating an embedding using PDD-weighted self attention can written like so:
> >
> > $X^{(1)} = X^{(0)} + \sigma \left( \frac{Q K^T}{\sqrt{d}} \right) V$
> >
> > where $X^{(0)} = R W_d$, $Q = X^{(0)}  W_Q$, $K = X_i^{(0)} W_K$ and $V = X^{(0)} W_{V}$. The matrices $R,W_V,W_K,W_Q$ are the same as in the original definition. The modifed version of the
> > softmax $\sigma$ is applied to each row $z$ in the argument and includes the PDD weights like so:
> >
> > $ \sigma(z)_i = (w_i e^{z_i}) / (\sum_j  w_j e^{z_j})$
> >
> > where $i$ and $j$ are used to index entries of $z$ or weight in the PDD weight vector.
> >
> > > However, I do think that the paper should fairly reflect that this continuity property of PDD is not used in this architecture
> >
> > Thank you for the feedback, use of this term will be more precise in the revision.
> >
> > Please let us know if anything mentioned is unclear.

---

> > > ### Comment · Reviewer_kc8a · 2023-11-16
> > >
> > > Thanks! With this corrected attention rule (but not with the one in the original manuscript), I can see that the network will be continuous. I'd be great if you could add a little proof of this property to the paper. I think that'll be instructive, and motivate the attention rule you propose.
> > >
> > > I agree with reviewer V7SM that important baselines are missing in experiments. Once these are added, I will reconsider my score.

---

> > > > ### Author Response · Authors · 2023-11-18
> > > >
> > > > We have added the proof and updated the comparison in the revision.
> > > >
> > > > Thank you and please let us know if you have any other feedback.

---

> > > > > ### Comment · Reviewer_kc8a · 2023-11-22
> > > > >
> > > > > I thank the authors for their responses. While my concern regarding continuity has been addressed, I maintain my score in light of the other reviewers' criticisms.

---

### Meta-Review · Area_Chair_SXmV · 2023-12-11

**Metareview:**

The paper presents a transformer model that incorporates the Pointwise Distance Distribution (PDD) and spatial encoding method for predicting material properties. Reviwers pointed out weaknesses such as suboptimal performance in predicting key electronic properties, lack of novelty in the k-nearest-neighbor construction, lack of details in the experiments, vague descriptions of the datasets, the necessity of understanding how isometry is encoded, lack of clarity, limited contribution, small-scale experiments, and poorly presented results, etc.

Many of those weaknesses have been well addressed during the discussion period. However, the paper still needs further improvements to convice reviewers, e.g., focusing more on formation energy in empirical studies, significant revision of the paper to incorporate the reviewers' comments, inclusion of more complex materials systems and surfaces, making evaluation more convincing, etc.

**Justification For Why Not Higher Score:**

While the authors have effectively addressed several concerns during the discussion phase, further enhancements are still needed to convince the reviewers. These improvements include giving more attention to formation energy in empirical studies, significantly revising the paper to incorporate reviewers' feedback, including more intricate material systems and surfaces, and providing more convincing evaluations.

**Justification For Why Not Lower Score:**

N/A

---

### Decision · Program_Chairs · 2024-01-16

Reject